# Optimal Learners for Realizable Regression: PAC Learning and Online Learning

**Idan Attias**
Ben-Gurion University of the Negev
`idanatti@post.bgu.ac.il`

**Steve Hanneke**
Purdue University
`steve.hanneke@gmail.com`

**Alkis Kalavasis**
Yale University
`alvertos.kalavasis@yale.edu`

**Amin Karbasi**
Yale University, Google Research
`amin.karbasi@yale.edu`

**Grigoris Velegkas**
Yale University
`grigoris.velegkas@yale.edu`

## Abstract

In this work, we aim to characterize the statistical complexity of realizable regression both in the PAC learning setting and the online learning setting.

Previous work had established the sufficiency of finiteness of the fat shattering dimension for PAC learnability and the necessity of finiteness of the scaled Natarajan dimension, but little progress had been made towards a more complete characterization since the work of Simon (SICOMP '97). To this end, we first introduce a minimax instance optimal learner for realizable regression and propose a novel dimension that both qualitatively and quantitatively characterizes which classes of real-valued predictors are learnable. We then identify a combinatorial dimension related to the Graph dimension that characterizes ERM learnability in the realizable setting. Finally, we establish a necessary condition for learnability based on a combinatorial dimension related to the DS dimension, and conjecture that it may also be sufficient in this context.

Additionally, in the context of online learning we provide a dimension that characterizes the minimax instance optimal cumulative loss up to a constant factor and design an optimal online learner for realizable regression, thus resolving an open question raised by Daskalakis and Golowich in STOC '22.

## 1 Introduction

Real-valued regression is one of the most fundamental and well-studied problems in statistics and data science [Vap99, GBC16, Bac21], with numerous applications in domains such as economics and medicine [DG17]. However, despite its significance and applicability, theoretical understanding of the statistical complexity of real-valued regression is still lacking.

Perhaps surprisingly, in the fundamental **realizable** Probably Approximately Correct (PAC) setting [Val84] and the **realizable** online setting [Lit88, DG22], we do not know of any characterizing dimension or optimal learners for the regression task. This comes in sharp contrast with binary and multiclass classification, both in the offline and the online settings, where the situation is much more clear [Lit88, HLW94, BEHW89, Han16, DSS14, BCD+22, Nat89, DSBDSS15, RST23]. Our goal in this work is to make progress regarding the following important question:

37th Conference on Neural Information Processing Systems (NeurIPS 2023).

*Which dimensions characterize PAC and online learnability for realizable real-valued regression?*

Consider an instance space $\mathcal{X}$, label space $\mathcal{Y} = [0, 1]$ and hypothesis class $\mathcal{H} \subseteq [0, 1]^{\mathcal{X}}$. In this work, we focus on the case of the regression framework with respect to the **absolute loss** $\ell(x, y) \triangleq |x - y|$ for any $x, y \in [0, 1]$, following previous works [BLW94, Sim97, ABDCBH97, BL98, DG22]. Our qualitative results hold for more general losses, namely any approximate pseudo-metric loss, which includes common losses such as $\ell_p$ for any $p \geq 1$. See the formal statements in Appendix H.

**PAC/Offline Realizable Regression.** Let us first recall the definition of realizable real-valued regression in the PAC setting. Informally, the learner is given i.i.d. labeled examples drawn from an unknown distribution $\mathcal{D}$ with the promise that there exists a target hypothesis $h^{\star} \in \mathcal{H}$ that perfectly labels the data. The goal is to use this training set to design a predictor with small error on future examples from the same distribution. Note that given a learner $A$ and sample $S \sim \mathcal{D}^n$, we let $A(S; x)$ be its prediction on $x \in \mathcal{X}$ when the training set is $S$.

**Definition 1** (PAC Realizable Regression). *Let $\ell : [0, 1]^2 \to \mathbb{R}_{\geq 0}$ be the absolute loss function. Consider a class $\mathcal{H} \subseteq [0, 1]^{\mathcal{X}}$ for some domain $\mathcal{X}$. Let $h^{\star} \in \mathcal{H}$ be an unknown target function and let $\mathcal{D}_{\mathcal{X}}$ be an unknown distribution on $\mathcal{X}$. A random sample $S$ of size $n$ consists of points $x_1, \ldots, x_n$ drawn i.i.d. from $\mathcal{D}_{\mathcal{X}}$ and the corresponding values $h^{\star}(x_1), \ldots, h^{\star}(x_n)$ of the target function.*

- *An algorithm $A : (\mathcal{X} \times [0, 1])^n \to [0, 1]^{\mathcal{X}}$ is an $n$-**sample PAC learner for $\mathcal{H}$ with respect to** $\ell$ if, for all $0 < \varepsilon, \delta < 1$, there exists $n = n(\varepsilon, \delta) \in \mathbb{N}$ such that for any $h^{\star} \in \mathcal{H}$ and any domain distribution $\mathcal{D}_{\mathcal{X}}$, it holds that $\mathbf{E}_{x \sim \mathcal{D}_{\mathcal{X}}}[\ell(A(S; x), h^{\star}(x))] \leq \varepsilon$, with probability at least $1 - \delta$ over $S \sim \mathcal{D}_{\mathcal{X}}^n$.*

- *An algorithm $A : (\mathcal{X} \times [0, 1])^n \to [0, 1]^{\mathcal{X}}$ is an $n$-**sample cut-off PAC learner for $\mathcal{H}$ with respect to** $\ell$ if, for all $0 < \varepsilon, \delta, \gamma < 1$, there exists $n = n(\varepsilon, \delta, \gamma) \in \mathbb{N}$ such that for any $h^{\star} \in \mathcal{H}$ and any domain distribution $\mathcal{D}_{\mathcal{X}}$, it holds that $\mathbf{Pr}_{x \sim \mathcal{D}_{\mathcal{X}}}[\ell(A(S; x), h^{\star}(x)) > \gamma] \leq \varepsilon$, with probability at least $1 - \delta$ over $S \sim \mathcal{D}_{\mathcal{X}}^n$.*

We remark that these two PAC learning definitions are qualitatively equivalent (cf. Lemma 1). We note that, throughout the paper, we implicitly assume, as e.g., in [HKS19], that all hypothesis classes are *admissible* in the sense that they satisfy mild measure-theoretic conditions, such as those specified in [DKLD84] (Section 10.3.1) or [Pol12] (Appendix C).

The question we would like to understand in this setting (and was raised by [Sim97]) follows:

**Question 1.** *Can we characterize learnability and design minimax optimal PAC learners for realizable regression?*

Traditionally, in the context of statistical learning a minimax optimal learner $A^{\star}$ is one that, for every class $\mathcal{H}$, given an error parameter $\varepsilon$ and a confidence parameter $\delta$ requires $\min_A \max_{\mathcal{D}} \mathcal{M}_A(\mathcal{H}; \varepsilon, \delta, \mathcal{D})$ samples to achieve it, where $\mathcal{M}_A(\mathcal{H}; \varepsilon, \delta, \mathcal{D})$ is the number of samples that some learner $A$ requires to achieve error $\varepsilon$ with confidence $\delta$ when the data-generating distribution is $\mathcal{D}$. There seems to be some inconsistency in the literature about realizable regression, where some works present minimax optimal learners when the maximum is also taken over the hypothesis class $\mathcal{H}$, i.e., $\min_A \max_{\mathcal{D}, \mathcal{H}} \mathcal{M}_A(\mathcal{H}, \varepsilon, \delta, \mathcal{D})$. This is a weaker result compared to ours, since it shows that there exists *some* hypothesis class for which these learners are optimal.

Our main results in this setting, together with the uniform convergence results from prior work, give rise to an interesting landscape of realizable PAC learning that is depicted in Figure 1.

**Online Realizable Regression.** We next shift our attention to the classical setting of online learning where the learner interacts with the adversary over a sequence of $T$ rounds: in every round the adversary presents an example $x_t \in \mathcal{X}$, the learner predicts a label $\widehat{y}_t$ and then the adversary reveals the correct label $y_t^{\star}$. In this context, realizability means that there always exists some function $h_t \in \mathcal{H}$ that perfectly explains the examples and the labels that the adversary has chosen. In the agnostic setting, the goal of the learner is to compete with the performance of the best function in $\mathcal{H}$, i.e., achieve small *regret*. This setting was introduced by [Lit88] in the context of binary classification, where they also characterized learnability in the realizable setting and provided an optimal algorithm. Later, [BDPSS09] provided an almost optimal algorithm in the agnostic setting, which suffered from an additional $\log T$ factor in its regret bound. This extra factor was later shaved

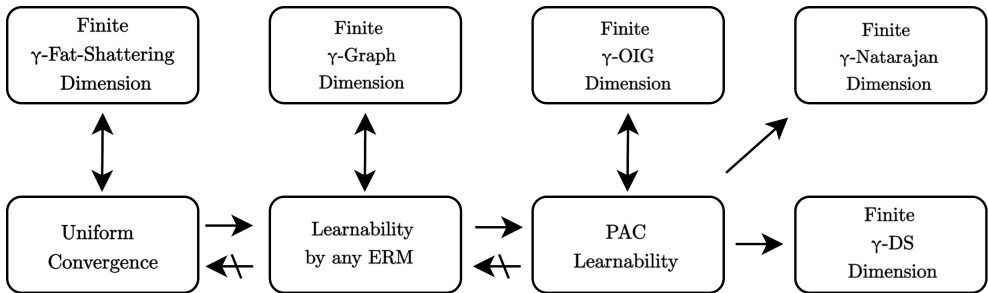

Figure 1: Landscape of Realizable PAC Regression: the "deleted" arrows mean that the implication is *not* true. The equivalence between finite fat-shattering dimension and the uniform convergence property is known even in the realizable case (see [SSSSS10]) and the fact that PAC learnability requires finite scaled Natarajan dimension is proved in [Sim97]. The properties of the other three dimensions (scaled Graph dimension, scaled One-Inclusion-Graph (OIG) dimension, and scaled Daniely-Shalev Shwartz (DS) dimension) are shown in this work. We further conjecture that finite scaled Natajaran dimension is not sufficient for PAC learning, while finite scaled DS does suffice. Interestingly, we observe that the notions of uniform convergence, learnability by any ERM and PAC learnability are separated in realizable regression.

by [ABED+21]. The study of multiclass online classification was initiated by [DSBDSS15], who provided an optimal algorithm for the realizable setting and an algorithm that is suboptimal by a factor of $\log k \cdot \log T$ in the agnostic setting, where $k$ is the total number of labels. Recently, [RST23] shaved off the $\log k$ factor. The problem of online regression differs significantly from that of online classification, since the loss function is not binary. The agnostic online regression setting has received a lot of attention and there is a series of works that provides optimal minimax guarantees [RST10, RS14, RST15b, RST15a, BDR21].

To the best of our knowledge, the realizable setting has received much less attention. A notable exception is the work of [DG22] that focuses on realizable online regression using *proper*[1] learners. They provide an optimal regret bound with respect to the *sequential fat-shattering dimension*. However, as they mention in their work (cf. Examples 1 and 2), this dimension does *not* characterize the optimal cumulative loss bound.

Interestingly, Daskalakis and Golowich [DG22] leave the question of providing a dimension that characterizes online realizable regression open. In our work, we resolve this question by providing upper and lower bounds for the cumulative loss of the learner that are tight up to a constant factor of 2 using a novel combinatorial dimension that is related to (scaled) Littlestone trees (cf. Definition 13). Formally, the setting of realizable online regression is defined as follows:

**Definition 2** (Online Realizable Regression). *Let $\ell : [0,1]^2 \to \mathbb{R}_{\geq 0}$ be a loss function. Consider a class $\mathcal{H} \subseteq [0,1]^{\mathcal{X}}$ for some domain $\mathcal{X}$. The realizable online regression setting over $T$ rounds consists of the following interaction between the learner and the adversary:*

- *The adversary presents $x_t \in \mathcal{X}$.*

- *The learner predicts $\widehat{y}_t \in [0,1]$, possibly using randomization.*

- *The adversary reveals the true label $y_t^\star \in [0,1]$ with the constraint that $\exists h_t^\star \in \mathcal{H}, \forall \tau \leq t, h(x_\tau) = y_\tau^\star$.*

- *The learner suffers loss $\ell(\widehat{y}_t, y_t^\star)$.*

*The goal of the learner is to minimize its expected cumulative loss $\mathfrak{C}_T = \mathbf{E}\left[\sum_{t \in [T]} \ell(\widehat{y}_t, y_t^\star)\right]$.*

We remark that in the definition of the cumulative loss $\mathfrak{C}_T$, the expectation is over the randomness of the algorithm (which is the only stochastic aspect of the online setting). As we explained before, the main question we study in this setting is the following:

---

[1] A learner is proper when the predictions $\widehat{y}_t$ can be realized by some function $h_t \in \mathcal{H}$.

**Question 2.** *Can we characterize the optimal cumulative loss and design optimal online learners for realizable regression?*

Our main result in this setting provides a dimension that characterizes the optimal cumulative loss up to a factor of 2 and provides an algorithm that achieves this bound.

## 1.1 Related Work

Our work makes progress towards the characterization of realizable regression both in the offline and online settings. Similar results are known for the more studied settings of binary and multiclass (PAC/online) classification [Lit88, HLW94, BEHW89, Han16, DSS14, BCD+22, Nat89, DSBDSS15, RBR09, BLM20, RST23, ABED+21, HKLM22, KKMV23, BGH+23]. The fundamental works of [BLW94, Sim97, ABDCBH97, BL98] study the question of PAC learnability for the regression task. However, none of them characterizes learnability in the realizable setting. [Sim97] showed that finiteness of scaled Natarajan dimension is necessary for realizable PAC regression. [BL98] employed the one-inclusion graph (OIG) algorithm to get a real-valued predictor whose expected error is upper bounded by the $V_\gamma$-dimension, whose finiteness is sufficient but not necessary for realizable learnability in this setting. We refer to [KS23] for details about this dimension which was introduced in [ABDCBH97]. [ABDCBH97] showed that finiteness of the fat shattering dimension at all scales is equivalent to PAC learnability in the agnostic setting. Nevertheless, this does not hold in the realizable case as Example 1 demonstrates. Similarly, the work of [BLW94] shows that fat shattering dimension characterizes the regression task when the labels are corrupted by noise. Recently, the concurrent and independent work of [AACSZ23] provides high-probability bounds for the one-inclusion graph algorithm in the realizable PAC regression setting using the $V_\gamma$-dimension. We underline that this dimension does not characterize realizable regression. For more general losses, see [Men02, BBM05].

In the area of online regression, the work of [DG22] studies the realizable setting of online regression with the absolute loss (as we do) and presents a randomized proper learning algorithm which achieves a near-optimal cumulative loss in terms of the sequential fat-shattering dimension of the hypothesis class. We emphasize that this dimension does not tightly capture this setting and, hence, does not address the question that we study. In particular, the lower bound they provide is related to the unimprovability of a bound concerning the sequential fat-shattering dimension and does not tightly capture the complexity of the problem. In the more general agnostic case, regret bounds have been obtained in the work of [RST15a], using the sequential fat-shattering dimension and sequential covering numbers. These quantities are also not tight in the realizable setting. See also [RS14, RST15b, RST17]. Moreover, regression oracles have been used in contextual bandits problems, see [FR20, SLX22] for further details.

Our characterization (cf. Theorem 2) for the offline setting via the OIG algorithm [HLW94] is further motivated by the work of [MHS22], where they propose a dimension of similar flavor for adversarially robust PAC learnability. We mention that recently the OIG algorithm has received a lot of attention from the statistical learning theory community [AACSZ22, BHM+21, KVK22, SMB22, AHHM22, CP22]. Finally, for a small sample of works that deal with offline and online regression problems, see [BDGR22, Gol21, AH22, HKS19, She17, MKFI22] and the references therein.

## 2 PAC Learnability for Realizable Regression

In this section, we present various combinatorial dimensions that provide necessary or sufficient conditions for learnability of real-valued functions. All the definitions that we consider have a similar flavor. We first define what it means for a class $\mathcal{H}$ to "shatter" a set of $n$ points and then we define the dimension to be equal to the cardinality of the largest set that $\mathcal{H}$ can shatter. Moreover, since we are considering real-valued learning, these dimensions are parameterized by a scaling factor $\gamma \in (0, 1)$ which should be interpreted as the distance that we can get to the optimal function. We start with the standard notion of projection of a class to a set of unlabeled examples.

**Definition 3** (Projection of $\mathcal{H}$ to $S$)**.** *Given $S = \{x_1, \ldots, x_n\} \in \mathcal{X}^n$, the projection of $\mathcal{H} \subseteq \mathcal{Y}^{\mathcal{X}}$ to $S$ is $\mathcal{H}|_S = \{(h(x_1), \ldots, h(x_n)) : h \in \mathcal{H}\}$.*

Furthermore, we say that a labeled sample $S \in (\mathcal{X} \times [0, 1])^n$ is realizable with respect to $\mathcal{H}$ if there exists $h \in \mathcal{H}$ such that $h(x_i) = y_i, \forall i \in [n]$.

## 2.1 $\gamma$-Fat Shattering Dimension

Perhaps the most well-known dimension in the real-valued learning setting is the fat shattering dimension that was introduced in [KS94]. Its definition is inspired by the pseudo-dimension [Pol90] and it is, essentially, a scaled version of it.

**Definition 4** ($\gamma$-Fat Shattering Dimension [KS94])**.** *Let $\mathcal{H} \subseteq [0,1]^{\mathcal{X}}$. We say that a sample $S \in \mathcal{X}^n$ is $\gamma$-fat shattered by $\mathcal{H}$ if there exist $s_1, \ldots, s_n \in [0,1]^n$ such that for all $b \in \{0,1\}^n$ there exists $h_b \in \mathcal{H}$ such that:*

- *$h_b(x_i) \geq s_i + \gamma, \forall i \in [n]$ such that $b_i = 1$.*

- *$h_b(x_i) \leq s_i - \gamma, \forall i \in [n]$ such that $b_i = 0$.*

*The $\gamma$-**fat shattering dimension** $\mathbb{D}_\gamma^{\mathrm{fat}}$ is defined to be the maximum size of a $\gamma$-fat shattered set.*

In the realizable setting, finiteness of the fat-shattering dimension (at all scales) is sufficient for learnability and it is equivalent to uniform convergence[2] [BLW94]. However, the next example shows that it is **not** a necessary condition for learnability. This comes in contrast to the agnostic case for real-valued functions, in which learnability and uniform convergence are equivalent for all $\mathcal{H}$.

**Example 1** (Realizable Learnability $\not\Rightarrow$ Finite Fat-Shattering Dimension, see Section 6 in [BLW94])**.** *Consider a class $\mathcal{H} \subseteq [0,1]^{\mathcal{X}}$ where each hypothesis is uniquely identifiable by a single example, i.e., for any $x \in \mathcal{X}$ and any $f, g \in \mathcal{H}$ we have that $f(x) \neq g(x)$, unless $f \equiv g$. Concretely, for every $d \in \mathbb{N}$, let $\{S_j\}_{0 \leq j \leq d-1}$ be a partition of $\mathcal{X}$ and define $\mathcal{H}_d = \{h_{b_0,\ldots,b_{d-1}} : b_i \in \{0,1\}, 0 \leq i \leq d-1\}$, where*

$$h_{b_0,\ldots,b_{d-1}}(x) = \frac{3}{4} \sum_{j=0}^{d-1} \mathbb{1}_{S_j}(x) b_j + \frac{1}{8} \sum_{k=0}^{d-1} b_k 2^{-k}.$$

*For any $\gamma \leq 1/4, \mathbb{D}_\gamma^{\mathrm{fat}}(\mathcal{H}_d) = d$, since for a set of points $x_0, \ldots, x_{d-1} \in \mathcal{X}$ such that each $x_j$ belongs to $S_j$, $0 \leq j \leq d-1$, $\mathcal{H}_d$ contains all possible patterns of values above $3/4$ and below $1/4$. Indeed, it is not hard to verify that for any $j \in \{0, \ldots, d-1\}$ if we consider any $h' := h_{b_0,\ldots,b_{d-1}} \in \mathcal{H}_d$ with $b_j = 1$ we have $h'(x_j) \geq 3/4$. Similarly, if $b_j = 0$ then it holds that $h'(x_j) \leq 1/4$. Hence, $\mathbb{D}_\gamma^{\mathrm{fat}} (\cup_{d \in \mathbb{N}} \mathcal{H}_d) = \infty$. Nevertheless, by just observing one example $(x, h^\star(x))$ any ERM learner finds the exact labeling function $h^\star$.*

We remark that this example already shows that the PAC learnability landscape of regression is quite different from that of multiclass classification, where agnostic learning and realizable learning are characterized by the same dimension [BCD+22].

To summarize this subsection, the fat-shattering dimension is a natural way to quantify how well the function class can interpolate (with gap $\gamma$) some fixed function. Crucially, this interpolation contains only inequalities (see Definition 4) and hence (at least intuitively) cannot be tight for the realizable setting, where there exists some function that exactly labels the features. Example 1 gives a natural example of a class with infinite fat-shattering dimension that can, nevertheless, be learned with a single sample in the realizable setting.

## 2.2 $\gamma$-Natarajan Dimension

The $\gamma$-Natarajan dimension was introduced by [Sim97] and is inspired by the Natarajan dimension [Nat89], which has been used to derive bounds in the multiclass classification setting. Before explaining the $\gamma$-Natarajan dimension, let us begin by reminding to the reader the standard Natarajan dimension. We say that a set $S = \{x_1, ..., x_n\}$ of size $n$ is Natarajan-shattered by a concept class $\mathcal{H} \subseteq \mathcal{Y}^{\mathcal{X}}$ if there exist two functions $f, g : S \to \mathcal{Y}$ so that $f(x_i) \neq g(x_i)$ for all $i \in [n]$, and for all $b \in \{0,1\}^n$ there exists $h \in \mathcal{H}$ such that $h(x_i) = f(x_i)$ if $b_i = 1$ and $h(x_i) = g(x_i)$ if $b_i = 0$. Note that here we have equalities instead of inequalities (recall the fat-shattering case Definition 4).

From a geometric perspective (see [BCD+22]), this means that the space $\mathcal{H}$ projected on the set $S$ contains the set $\{f(x_1), g(x_1)\} \times ... \times \{f(x_n), g(x_n)\}$. This set is "isomorphic" to the Boolean

---

[2]Informally, uniform convergence means that for all distributions $\mathcal{D}$, with high probability over the sample, the error of all $h \in \mathcal{H}$ on the sample is close to their true population error.

hypercube of size $n$ by mapping $f(x_i)$ to 1 and $g(x_i)$ to 0 for all $i \in [n]$. This means that the Natarajan dimension is essentially the size of the largest Boolean cube contained in $\mathcal{H}$.

[Sim97] defines the scaled analogue of the above dimension as follows.

**Definition 5** ($\gamma$-Natarajan Dimension [Sim97])**.** *Let $\mathcal{H} \subseteq [0,1]^{\mathcal{X}}$. We say that a set $S \in \mathcal{X}^n$ is $\gamma$-Natarajan-shattered by $\mathcal{H}$ if there exist $f, g : [n] \to [0,1]$ such that for every $i \in [n]$ we have $\ell(f(i), g(i)) \geq 2\gamma$, and*

$$\mathcal{H}|_S \supseteq \{f(1), g(1)\} \times \ldots \times \{f(n), g(n)\}.$$

*The $\gamma$-**Natarajan dimension** $\mathbb{D}_{\gamma}^{\mathrm{Nat}}$ is defined to be the maximum size of a $\gamma$-Natarajan-shattered set.*

Intuitively, one should think of the $\gamma$-Natarajan dimension as indicating the size of the largest Boolean cube that is contained in $\mathcal{H}$. Essentially, every coordinate $i \in [n]$ gets its own translation of the $0, 1$ labels of the Boolean cube, with the requirement that these two labels are at least $2\gamma$ far from each other. [Sim97] showed the following result, which states that finiteness of the Natarajan dimension at all scales is a necessary condition for realizable PAC regression:

**Informal Theorem 1** (Theorem 3.1 in [Sim97])**.** *$\mathcal{H} \subseteq [0,1]^{\mathcal{X}}$ is PAC learnable in the realizable regression setting only if $\mathbb{D}_{\gamma}^{\mathrm{Nat}}(\mathcal{H}) < \infty$ for all $\gamma \in (0,1)$.*

Concluding these two subsections, we have explained the main known general results about realizable offline regression: (i) finiteness of fat-shattering is sufficient but not necessary for PAC learning and (ii) finiteness of scaled Natarajan is necessary for PAC learning.

### 2.3 $\gamma$-Graph Dimension

We are now ready to introduce the $\gamma$-graph dimension, which is a relaxation of the definition of the $\gamma$-Natarajan dimension. To the best of our knowledge, it has not appeared in the literature before. Its definition is inspired by its non-scaled analogue in multiclass classification [Nat89, DSS14].

**Definition 6** ($\gamma$-Graph Dimension)**.** *Let $\mathcal{H} \subseteq [0,1]^{\mathcal{X}}, \ell : \mathbb{R}^2 \to [0,1]$. We say that a sample $S \in \mathcal{X}^n$ is $\gamma$-graph shattered by $\mathcal{H}$ if there exists $f : [n] :\to [0,1]$ such that for all $b \in \{0,1\}^n$ there exists $h_b \in \mathcal{H}$ such that:*

- *$h_b(x_i) = f(i), \forall i \in [n]$ such that $b_i = 0$.*
- *$\ell(h_b(x_i), f(i)) > \gamma, \forall i \in [n]$ such that $b_i = 1$.*

*The $\gamma$-**graph dimension** $\mathbb{D}_{\gamma}^{\mathrm{G}}$ is defined to be the maximum size of a $\gamma$-graph shattered set.*

We mention that the asymmetry in the above definition is crucial. In particular, replacing the equality $h_b(x_i) = f(i)$ with $\ell(h_b(x_i), f(i)) \leq \gamma$ fails to capture the properties of the graph dimension. Intuitively, the equality in the definition reflects the assumption of realizability, i.e., the guarantee that there exists a hypothesis $h^\star$ that exactly fits the labels. Before stating our main result, we can collect some useful observations about this new combinatorial measure. In particular, we provide examples inspired by [DSS14, DSBDSS15] which show (i) that there exist gaps between different ERM learners (see Example 3) and (ii) that any learning algorithm with a close to optimal sample complexity must be improper (see Example 4).

Our first main result relates the scaled graph dimension with the learnability of any class $\mathcal{H} \subseteq [0,1]^{\mathcal{X}}$ using a (worst case) ERM learner. This result is the scaled analogue of known multiclass results [DSS14, DSBDSS15] but its proof for the upper bound follows a different path. For the formal statement of our result and its full proof, we refer the reader to Appendix B.

**Informal Theorem 2** (Informal, see Theorem 1)**.** *Any $\mathcal{H} \subseteq [0,1]^{\mathcal{X}}$ is PAC learnable in the realizable regression setting by a worst-case ERM learner if and only if $\mathbb{D}_{\gamma}^{\mathrm{G}}(\mathcal{H}) < \infty$ for all $\gamma \in (0,1)$.*

**Proof Sketch.** The proof of the lower bound follows in a similar way as the lower bound regarding learnability of binary hypothesis classes that have infinite VC dimension [VC71, BEHW89]. If $\mathcal{H}$ has infinite $\gamma$-graph dimension for some $\gamma \in (0,1)$, then for any $n \in \mathbb{N}$ we can find a sequence of $n$ points $x_1, \ldots, x_n$ that are $\gamma$-graph shattered by $\mathcal{H}$. Then, we can define a distribution $\mathcal{D}$ that puts most of its mass on $x_1$, so if the learner observes $n$ samples then with high probability it will not observe at least half of the shattered points. By the definition of the $\gamma$-graph dimension this shows

that, with high probability, there exists at least one ERM learner that is $\gamma$-far on at least half of these points. The result follows by taking $n \to \infty$.

The most technically challenging part of our result is the upper bound. There are three main steps in our approach. First, we argue that by introducing a "ghost" sample of size $n$ on top of the training sample, which is also of size $n$, we can bound the true error of any ERM learner on the distribution by twice its error on the ghost sample. This requires a slightly more subtle treatment compared to the argument of [BEHW89] for binary classification. The second step is to use a "random swaps" type of argument in order to bound the performance of the ERM learner on the ghost sample as follows: we "merge" these two samples by considering any realizable sequence $S$ of $2n$ points, we randomly swap elements whose indices differ by $n$, and then we consider an ERM learner who gets trained on the first $n$ points. We can show that if such a learner does not make many mistakes on the unseen part of the sequence, in expectation over the random swaps, then it has a small error on the true distribution. The main advantage of this argument is that it allows us to bound the number of mistakes of such a learner on the unseen part of the sequence without using any information about $\mathcal{D}$. The last step of the proof is where we diverge from the argument of [BEHW89]. In order to show that this expectation is small, we first map $\mathcal{H}$ to a *partial concept class*[3] $\overline{\mathcal{H}}$ [AHHM22], then we project $\overline{\mathcal{H}}$ to the sequence $S$, and finally we map $\overline{\mathcal{H}}$ back to a total concept class by considering a *disambiguation* of it. Through these steps, we can show that there will be at most $(2n)^{O(\mathbb{D}_\gamma^G(\mathcal{H})\log(2n))}$ many different functions in this "projected" class. Then, we can argue that, with high probability, any ERM learner who sees the first half of the sequence makes, in expectation over the random swaps, a small number of mistakes on the second half of the sequence. Finally, we can take a union bound over all the $(2n)^{O(\mathbb{D}_\gamma^G(\mathcal{H})\log(2n))}$ possible such learners to conclude the proof.

## 2.4 $\gamma$-One-Inclusion Graph Dimension

In this section, we provide a minimax optimal learner for realizable PAC regression. We first review a fundamental construction which is a crucial ingredient in the design of our learner, namely the one-inclusion (hyper)graph (OIG) algorithm $\mathbb{A}^{\mathrm{OIG}}$ for a class $\mathcal{H} \subseteq \mathcal{Y}^{\mathcal{X}}$ [HLW94, RBR09, DSS14, BCD$^+$22]. This algorithm gets as input a training set $(x_1, y_1), ..., (x_n, y_n)$ realizable by $\mathcal{H}$ and an additional example $x$. The goal is to predict the label of $x$. Let $S = \{x_1, ..., x_n, x\}$. The idea is to construct the one-inclusion graph $G_{\mathcal{H}|_S}^{\mathrm{OIG}}$ induced by the pair $(S, \mathcal{H})$. The node set $V$ of this graph corresponds to the set $\mathcal{H}|_S$ (projection of $\mathcal{H}$ to $S$) and, so, $V \subseteq \mathcal{Y}^{[n+1]}$. For the binary classification case, two vertices are connected with an edge if they differ in exactly one element $x$ of the $n+1$ points in $S$. For the case where $\mathcal{Y}$ is discrete and $|\mathcal{Y}| > 2$, the hyperedge is generalized accordingly.

**Definition 7** (One-Inclusion Hypergraph [HLW94, RBR09, BCD$^+$22]). *Consider the set $[n]$ and a hypothesis class $\mathcal{H} \subseteq \mathcal{Y}^{[n]}$. We define a graph $G_{\mathcal{H}}^{\mathrm{OIG}} = (V, E)$ such that $V = \mathcal{H}$. Consider a direction $i \in [n]$ and a mapping $f : [n] \setminus \{i\} \to \mathcal{Y}$. We introduce the hyperedge $e_{i,f} = \{h \in V : h(j) = f(j), \ \forall j \in [n] \setminus \{i\}\}$. We define the edge set of $G_{\mathcal{H}}^{\mathrm{OIG}}$ to be the collection*

$$E = \{e_{i,f} : i \in [n], f : [n] \setminus \{i\} \to \mathcal{Y}, e_{i,f} \neq \emptyset\}.$$

In the regression setting, having created the one-inclusion graph with $\mathcal{Y} = [0, 1]$, the goal is to *orient* the edges; the crucial property is that "good" orientations of this graph yield learning algorithms with low error. An orientation is good if the maximum out-degree of the graph is small (cf. Definition 8). Informally, if the maximum out-degree of any node is $M$, then we can create a predictor whose expected error rate is at most $M/(n+1)$. Note that in the above discussion we have not addressed the issue that we deal with regression tasks and not classification and, hence, some notion of *scale* is required in the definition of the out-degree.

Intuitively, the set $S$ that induces the vertices $\mathcal{H}|_S$ of the OIG consists of the features of the training examples $\{x_1, ..., x_n\}$ and the test point $x$. Hence, each edge of the OIG should be thought of as the set of all potential labels of the test point that are realizable by $\mathcal{H}$, so it corresponds to the set of all possible meaningful predictions for $x$. An orientation $\sigma$ maps every edge $e$ to a vertex $v \in e$ and, hence, it is equivalent to the prediction of a learning algorithm. We can now formally define the notion of an orientation and the scaled out-degree.

---

[3] For an introduction to partial concept classes and their disambiguations we refer the reader to Appendix B.1.

**Definition 8** (Orientation and Scaled Out-Degree). *Let $\gamma \in [0,1], n \in \mathbb{N}, \mathcal{H} \subseteq [0,1]^{[n]}$. An orientation of the one-inclusion graph $G_{\mathcal{H}}^{\text{OIG}} = (V, E)$ is a mapping $\sigma : E \to V$ so that $\sigma(e) \in e$ for any $e \in E$. Let $\sigma_i(e) \in [0,1]$ denote the $i$-th entry of the orientation.*

*For a vertex $v \in V$, corresponding to some hypothesis $h \in \mathcal{H}$ (see Definition 7), let $v_i$ be the $i$-th entry of $v$, which corresponds to $h(i)$. The (scaled) out-degree of a vertex $v$ under $\sigma$ is $\text{outdeg}(v; \sigma, \gamma) = |\{i \in [n] : \ell(\sigma_i(e_{i,v}), v_i) > \gamma\}|$. The maximum (scaled) out-degree of $\sigma$ is $\text{outdeg}(\sigma, \gamma) = \max_{v \in V} \text{outdeg}(v; \sigma, \gamma)$.*

Finally, we introduce the following novel dimension in the context of real-valued regression. An analogous dimension was proposed in the context of learning under adversarial robustness [MHS22].

**Definition 9** ($\gamma$-OIG Dimension). *Consider a class $\mathcal{H} \subseteq [0,1]^{\mathcal{X}}$ and let $\gamma \in [0,1]$. We define the $\gamma$-one-inclusion graph dimension $\mathbb{D}_{\gamma}^{\text{OIG}}$ of $\mathcal{H}$ as follows:*

$$\mathbb{D}_{\gamma}^{\text{OIG}}(\mathcal{H}) = \sup\{n \in \mathbb{N} : \exists S \in \mathcal{X}^n \text{ such that } \exists \text{ finite subgraph } G = (V, E) \text{ of } G_{\mathcal{H}|_S}^{\text{OIG}} = (V_n, E_n)$$
$$\text{such that } \forall \text{ orientations } \sigma, \exists v \in V, \text{ where } \text{outdeg}(v; \sigma, \gamma) > n/3\}.$$

*We define the dimension to be infinite if the supremum is not attained by a finite $n$.*

In words, it is the largest $n \in \mathbb{N}$ (potentially $\infty$) such that there exists an (unlabeled) sequence $S$ of length $n$ with the property that no matter how one orients some finite subgraph of the one-inclusion graph, there is always some vertex for which at least $1/3$ of its coordinates are $\gamma$-different from the labels of the edges that are attached to this vertex. We remark that the hypothesis class in Example 1 has $\gamma$-OIG dimension equal to $O(1)$. Moreover, a finite fat-shattering dimension of hypothesis class implies a finite OIG dimension of roughly the same size (see Appendix C). We also mention that the above dimension satisfies the "finite character" property and the remaining criteria that dimensions should satisfy according to [BDHM$^+$19] (see Appendix F). As our main result in this section, we show that any class $\mathcal{H}$ is learnable if and only if this dimension is finite and we design an (almost) optimal learner for it.

**Informal Theorem 3** (Informal, see Theorem 2). *Any $\mathcal{H} \subseteq [0,1]^{\mathcal{X}}$ is PAC learnable in the realizable regression setting if and only if $\mathbb{D}_{\gamma}^{\text{OIG}}(\mathcal{H}) < \infty$ for all $\gamma \in (0,1)$.*

The formal statement and its full proof are postponed to Appendix C.

**Proof Sketch.** We start with the lower bound. As we explained before, orientations of the one-inclusion graph are, in some sense, equivalent to learning algorithms. Therefore, if this dimension is infinite for some $\gamma > 0$, then for any $n \in \mathbb{N}$, there are no orientations with small maximum out-degree. Thus, for *any* learner we can construct *some* distribution $\mathcal{D}$ under which it makes a prediction that is $\gamma$-far from the correct one with constant probability, which means that $\mathcal{H}$ is not PAC learnable.

Let us now describe the proof of the converse direction, which consists of several steps. First, notice that the finiteness of this dimension provides good orientations for *finite* subgraphs of the one-inclusion graph. Using the compactness theorem of first-order logic, we can extend them to a good orientation of the whole, potentially infinite, one-inclusion graph. This step gives us a weak learner with the following property: for any given $\gamma$ there is some $n_0 \in \mathbb{N}$ so that, with high probability over the training set, when it is given $n_0$ examples as its training set it makes mistakes that are of order at least $\gamma$ on a randomly drawn point from $\mathcal{D}$ with probability at most $1/3$. The next step is to boost the performance of this weak learner. This is done using the "median-boosting" technique [Kég03] (cf. Algorithm 2) which guarantees that after a small number of iterations we can create an ensemble of weak learners such that, a prediction rule according to their (weighted) median will not make any $\gamma$-mistakes on the training set. However, this is not sufficient to prove that the ensemble of these learners has small loss on the distribution $\mathcal{D}$. This is done by establishing *sample compression* schemes that have small length. Essentially, such schemes consist of a *compression* function $\kappa$ which takes as input a training set and outputs a subset of it, and a reconstruction function $\rho$ which takes as input the output of $\kappa$ and returns a predictor whose error on every point of the training set $S$ is at most $\gamma$. Extending the arguments of [LW86] from the binary setting to the real-valued setting we show that the existence of such a scheme whose compression function returns a set of "small" cardinality implies generalization properties of the underlying learning rule. Finally, we show that our weak learner combined with the boosting procedure admit such a sample compression scheme.

Before proceeding to the next section, one could naturally ask whether there is a natural property of the concept class that implies finiteness of the scaled OIG dimension. The work of [Men02] provides

a sufficient and natural condition that implies finiteness of our complexity measure. In particular, Mendelson shows that classes that contain functions with bounded oscillation (as defined in [Men02]) have finite fat-shattering dimension. This implies that the class is learnable in the agnostic setting and hence is also learnable in the realizable setting. As a result, the OIG-based dimension is also finite. So, bounded oscillations are a general property that guarantees that the finiteness of OIG-based dimension and fat-shattering dimension coincide. We also mention that deriving bounds for the OIG-dimension for interesting families of functions is an important yet non-trivial question.

## 2.5  $\gamma$-DS Dimension

So far we have identified a dimension (cf. Definition 9) that characterizes the PAC learnability of realizable regression. However, it might not be easy to calculate it in some settings. Our goal in this section is to introduce a relaxation of this definition which we conjecture that also characterizes learnability in this context. This new dimension is inspired by the DS dimension, a combinatorial dimension defined by Daniely and Shalev-Shwartz in [DSS14]. In a recent breakthrough result, [BCD+22] showed that the DS dimension characterizes multiclass learnability (with a possibly unbounded number of labels and the 0-1 loss). We introduce a scaled version of the DS dimension. To this end, we first define the notion of a scaled pseudo-cube.

**Definition 10** (Scaled Pseudo-Cube). *Let $\gamma \in [0, 1]$. A class $\mathcal{H} \subseteq [0, 1]^d$ is called a $\gamma$-**pseudo-cube** of dimension $d$ if it is non-empty, finite and, for any $f \in \mathcal{H}$ and direction $i \in [d]$, the hyper-edge $e_{i,f} = \{g \in \mathcal{H} : g(j) = f(j) \ \ \forall j \in [d], i \neq j\}$ satisfies $|e_{i,f}| > 1$ and $\ell(g_1(i), g_2(i)) > \gamma$ for any $g_1, g_2 \in e_{i,f}, g_1 \neq g_2$.*

Pseudo-cubes can be seen as a relaxation of the notion of a Boolean cube (which should be intuitively related with the Natarajan dimension) and were a crucial tool in the proof of [BCD+22]. In our setting, scaled pseudo-cubes will give us the following combinatorial dimension.

**Definition 11** ($\gamma$-DS Dimension). *Let $\mathcal{H} \subseteq [0, 1]^{\mathcal{X}}$. A set $S \in \mathcal{X}^n$ is $\gamma$-DS shattered if $\mathcal{H}|_S$ contains an $n$-dimensional $\gamma$-pseudo-cube. The $\gamma$-DS **dimension** $\mathbb{D}_\gamma^{\mathrm{DS}}$ is the maximum size of a $\gamma$-DS-shattered set.*

Extending the ideas from the multiclass classification setting, we show that the scaled-DS dimension is necessary for realizable PAC regression. Simon (Section 6, [Sim97]) left as an open direction to "obtain supplementary lower bounds [for realizable regression] (perhaps completely unrelated to the combinatorial or Natarajan dimension)". Our next result is a novel contribution to this direction.

**Informal Theorem 4** (Informal, see Theorem 3). *Any $\mathcal{H} \subseteq [0, 1]^{\mathcal{X}}$ is PAC learnable in the realizable regression setting only if $\mathbb{D}_\gamma^{\mathrm{DS}}(\mathcal{H}) < \infty$ for all $\gamma \in (0, 1)$.*

The proof is postponed to Appendix D. We believe that finiteness of $\mathbb{D}_\gamma^{\mathrm{DS}}(\mathcal{H})$ is also a *sufficient* condition for realizable regression. However, the approach of [BCD+22] that establishes a similar result in the setting of multiclass classification does not extend trivially to the regression setting.

**Conjecture 1** (Finite $\gamma$-DS is Sufficient). *Let $\mathcal{H} \subseteq [0, 1]^{\mathcal{X}}$. If $\mathbb{D}_\gamma^{\mathrm{DS}}(\mathcal{H}) < \infty$ for all $\gamma \in (0, 1)$, then $\mathcal{H}$ is PAC learnable in the realizable regression setting.*

## 3  Online Learnability for Realizable Regression

In this section we will provide our main result regarding realizable online regression. Littlestone trees have been the workhorse of online classification problems [Lit88]. First, we provide a definition for scaled Littlestone trees.

**Definition 12** (Scaled Littlestone Tree). *A scaled Littlestone tree of depth $d \leq \infty$ is a complete binary tree of depth $d$ defined as a collection of nodes*

$$\bigcup_{0 \leq \ell < d} \{x_u \in \mathcal{X} : u \in \{0, 1\}^\ell\} = \{x_\emptyset\} \cup \{x_0, x_1\} \cup \{x_{00}, x_{01}, x_{10}, x_{11}\} \cup ...$$

*and real-valued gaps*

$$\bigcup_{0 \leq \ell < d} \{\gamma_u \in [0, 1] : u \in \{0, 1\}^\ell\} = \{\gamma_\emptyset\} \cup \{\gamma_0, \gamma_1\} \cup \{\gamma_{00}, \gamma_{01}, \gamma_{10}, \gamma_{11}\} \cup ...$$

*such that for every path $\boldsymbol{y} \in \{0,1\}^d$ and finite $n < d$, there exists $h \in \mathcal{H}$ so that $h(x_{\boldsymbol{y}_{\leq \ell}}) = s_{\boldsymbol{y}_{\leq \ell+1}}$ for $0 \leq \ell \leq n$, where $s_{\boldsymbol{y}_{\leq \ell+1}}$ is the label of the edge connecting the nodes $x_{\boldsymbol{y}_{\leq \ell}}$ and $x_{\boldsymbol{y}_{\leq \ell+1}}$ and $\ell\left(s_{\boldsymbol{y}_{\leq \ell},0}, s_{\boldsymbol{y}_{\leq \ell},1}\right) = \gamma_{\boldsymbol{y}_{\leq \ell}}$.*

In words, scaled Littlestone trees are complete binary trees whose nodes are points of $\mathcal{X}$ and the two edges attached to every node are its potential classifications. An important quantity is the *gap* between the two values of the edges. We define the online dimension $\mathbb{D}^{\mathrm{onl}}(\mathcal{H})$ as follows.

**Definition 13** (Online Dimension). *Let $\mathcal{H} \subseteq [0,1]^{\mathcal{X}}$. Let $\mathcal{T}_d$ be the space of all scaled Littlestone trees of depth $d$ (cf. Definition 12) and $\mathcal{T} = \bigcup_{d=0}^{\infty} \mathcal{T}_d$. For any scaled tree $T = \bigcup_{0 \leq \ell \leq \mathrm{dep}(T)} \left\{ (x_u, \gamma_u) \in (\mathcal{X}, [0,1]) : u \in \{0,1\}^{\ell} \right\}$, let $\mathcal{P}(T) = \{y = (y_0, ..., y_{\mathrm{dep}(T)}) : y_i \in \{0,1\}^i)\}$ be the set of all paths in $T$. The dimension $\mathbb{D}^{\mathrm{onl}}(\mathcal{H})$ is*

$$\mathbb{D}^{\mathrm{onl}}(\mathcal{H}) = \sup_{T \in \mathcal{T}} \inf_{y \in \mathcal{P}(T)} \sum_{i=0}^{\mathrm{dep}(T)} \gamma_{y_i}. \tag{1}$$

In words, this dimension considers the tree that has the maximum sum of label gaps over its path with the smallest such sum, among all the trees of arbitrary depth. Note that we are taking the supremum (infimum) in case there is no tree (path) that achieves the optimal value. Providing a characterization and a learner with optimal cumulative loss $\mathfrak{C}_T$ for realizable online regression was left as an open problem by [DG22]. We resolve this question (up to a factor of 2) by showing the following result.

**Informal Theorem 5** (Informal, see Theorem 4). *For any $\mathcal{H} \subseteq [0,1]^{\mathcal{X}}$ and $\varepsilon > 0$, there exists a deterministic learner with $\mathfrak{C}_{\infty} \leq \mathbb{D}^{\mathrm{onl}}(\mathcal{H}) + \varepsilon$ and any, potentially randomized, learner satisfies $\mathfrak{C}_{\infty} \geq \mathbb{D}^{\mathrm{onl}}(\mathcal{H})/2 - \varepsilon$.*

The formal statement and the full proof of our results can be found in Appendix E. First, we underline that this result holds when there is no bound on the number of rounds that the learner and the adversary interact. This follows the same spirit as the results in the realizable binary and multiclass classification settings [Lit88, DSBDSS15]. The dimension that characterizes the minimax optimal cumulative loss for any given $T$ follows by taking the supremum in Definition 13 over trees whose depth is at most $T$ and the proof follows in an identical way (note that even with finite fixed depth $T$ the supremum is over infinitely many trees). Let us now give a sketch of our proofs.

**Proof Sketch.** The lower bound follows using similar arguments as in the classification setting: for any $\varepsilon > 0$, the adversary can create a scaled Littlestone tree $T$ that achieves the sup inf bound, up to an additive $\varepsilon$. In the first round, the adversary presents the root of the tree $x_{\emptyset}$. Then, no matter what the learner picks the adversary can force error at least $\gamma_{\emptyset}/2$. The game is repeated on the new subtree. The proof of the upper bound presents the main technical challenge to establish Theorem 4. The strategy of the learner can be found in Algorithm 3. The key insight in the proof is that, due to realizability, we can show that in every round $t$ there is some $\widehat{y}_t \in [0,1]$ the learner can predict so that, no matter what the adversary picks as the true label $y_t^{\star}$, the online dimension of the class under the extra restriction that $h(x_t) = y_t^{\star}$, i.e, the updated *version space* $V = \{h \in \mathcal{H} : h(x_{\tau}) = y_{\tau}^{\star}, 1 \leq \tau \leq t\}$, will decrease by $\ell(y_t^{\star}, \widehat{y}_t)$. Thus, under this strategy of the learner, the adversary can only distribute up to $\mathbb{D}^{\mathrm{onl}}(\mathcal{H})$ across all the rounds of the interaction. We explain how the learner can find such a $\widehat{y}_t$ and we handle technical issues that arise due to the fact that we are dealing with sup inf instead of max min in the formal proof (cf. Appendix E).

## 4 Conclusion

In this work, we developed optimal learners for realizable regression in PAC learning and online learning. Moreover, we identified combinatorial dimensions that characterize learnability in these settings. We hope that our work can lead to simplified characterizations for these problems. We believe that the main limitation of our work is that the OIG-based dimension we propose is more complicated than the dimensions that have been proposed in the past, like the fat-shattering dimension (which, as we explain, does not characterize learnability in the realizable regression setting). Nevertheless, despite its complexity, this is the first dimension that characterizes learnability in the realizable regression setting. More to that, our work leaves as an important next step to prove (or disprove) the conjecture that the (combinatorial and simpler) $\gamma$-DS dimension is qualitatively equivalent to the $\gamma$-OIG dimension. Another future direction, that is not directly related to this conjecture, is to better understand the gap between the fat-shattering dimension and the OIG-based dimension.

## Acknowledgements

Amin Karbasi acknowledges funding in direct support of this work from NSF (IIS-1845032), ONR (N00014- 19-1-2406), and the AI Institute for Learning-Enabled Optimization at Scale (TILOS). Grigoris Velegkas is supported by TILOS, the Onassis Foundation, and the Bodossaki Foundation. This work was done in part while some of the authors were visiting Archimedes AI Research Center.

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

## A    Notation and Definitions

We first overview standard definitions about sample complexity in PAC learning.

**PAC Sample Complexity.**    The realizable PAC sample complexity $\mathcal{M}(\mathcal{H}; \varepsilon, \delta)$ of $\mathcal{H}$ is defined as

$$\mathcal{M}(\mathcal{H}; \varepsilon, \delta) = \inf_{A \in \mathcal{A}} \mathcal{M}_A(\mathcal{H}; \varepsilon, \delta), \tag{1}$$

where the infimum is over all possible learning algorithms and $\mathcal{M}_A(\mathcal{H}; \varepsilon, \delta)$ is the minimal integer such that for any $m \geq \mathcal{M}_A(\mathcal{H}; \varepsilon, \delta)$, every distribution $\mathcal{D}_\mathcal{X}$ on $\mathcal{X}$, and, true target $h^\star \in \mathcal{H}$, the expected loss $\mathbf{E}_{x \sim \mathcal{D}_\mathcal{X}}[\ell(A(T)(x), h^\star(x))]$ of $A$ is at most $\varepsilon$ with probability $1 - \delta$ over the training set $T = \{(x, h^\star(x)) : x \in S\}, S \sim \mathcal{D}_\mathcal{X}^m$.

**PAC Cut-Off Sample Complexity.**    We slightly overload the notation of the sample complexity and we define

$$\mathcal{M}(\mathcal{H}; \varepsilon, \delta, \gamma) = \inf_{A \in \mathcal{A}} \mathcal{M}_A(\mathcal{H}; \varepsilon, \delta, \gamma), \tag{2}$$

where the infimum is over all possible learning algorithms and $\mathcal{M}_A(\mathcal{H}; \varepsilon, \delta, \gamma)$ is the minimal integer such that for any $m \geq \mathcal{M}_A(\mathcal{H}; \varepsilon, \delta, \gamma)$, every distribution $\mathcal{D}_\mathcal{X}$ on $\mathcal{X}$, and, true target $h^\star \in \mathcal{H}$, the expected cut-off loss $\mathbf{Pr}_{x \sim \mathcal{D}_\mathcal{X}}[\ell(A(T)(x), h^\star(x)) > \gamma]$ of $A$ is at most $\varepsilon$ with probability $1 - \delta$ over the training set $T = \{(x, h^\star(x)) : x \in S\}, S \sim \mathcal{D}_\mathcal{X}^m$.

**Lemma 1** (Equivalence Between Sample Complexities)**.**    *For every $\varepsilon, \delta \in (0, 1)^2$ and every $\mathcal{H} \subseteq [0, 1]^\mathcal{X}$, where $\mathcal{X}$ is the input domain, it holds that*

$$\mathcal{M}(\mathcal{H}; \sqrt{\varepsilon}, \delta, \sqrt{\varepsilon}) \leq \mathcal{M}(\mathcal{H}; \varepsilon, \delta) \leq \mathcal{M}(\mathcal{H}; \varepsilon/2, \delta, \varepsilon/2)$$

*Proof.*  Let $A$ be a learning algorithm. We will prove the statement for each fixed $A$ and for each data-generating distribution $\mathcal{D}$, so the result follows by taking the infimum over the learning algorithms.

Assume that the cut-off sample complexity of $A$ is $\mathcal{M}_A(\mathcal{H}; \varepsilon/2, \delta, \varepsilon/2)$. Then, with probability $1 - \delta$ over the training sample $S \sim \mathcal{D}$, for its expected loss it holds that

$$\mathop{\mathbf{E}}_{x \sim \mathcal{D}_\mathcal{X}}[\ell(A(S; x), h^\star(x))] \leq \frac{\varepsilon}{2} + \left(1 - \frac{\varepsilon}{2}\right) \cdot \frac{\varepsilon}{2} \leq \varepsilon,$$

thus, $\mathcal{M}_A(\mathcal{H}; \varepsilon, \delta) \leq \mathcal{M}_A(\mathcal{H}; \varepsilon/2, \delta, \varepsilon/2)$.

The other direction follows by using Markov's inequality. In particular, if we have that with probability at least $1 - \delta$ over $S \sim \mathcal{D}$ it holds that

$$\mathop{\mathbf{E}}_{x \sim \mathcal{D}_\mathcal{X}}[\ell(A(S; x), h^\star(x))] \leq \varepsilon,$$

then Markov's inequality gives us that

$$\mathop{\mathbf{Pr}}_{x \sim \mathcal{D}_\mathcal{X}}[\ell(A(S; x), h^\star(x)) \geq \sqrt{\varepsilon}] \leq \sqrt{\varepsilon},$$

which shows that $\mathcal{M}_A(\mathcal{H}; \varepsilon, \delta) \geq \mathcal{M}_A(\mathcal{H}; \sqrt{\varepsilon}, \delta, \sqrt{\varepsilon})$.    $\square$

**ERM Sample Complexity.**    In the special case where $\mathcal{A}$ is the class $\mathrm{ERM}$ of all possible ERM algorithms, i.e., algorithms that return a hypothesis whose sample error is exactly 0, we define the ERM sample complexity as the number of samples required by the worst-case ERM algorithm, i.e.,

$$\mathcal{M}_{\mathrm{ERM}}(\mathcal{H}; \varepsilon, \delta) = \sup_{A \in \mathrm{ERM}} \mathcal{M}_A(\mathcal{H}; \varepsilon, \delta), \tag{3}$$

and its cut-off analogue as

$$\mathcal{M}_{\mathrm{ERM}}(\mathcal{H}; \varepsilon, \delta, \gamma) = \sup_{A \in \mathrm{ERM}} \mathcal{M}_A(\mathcal{H}; \varepsilon, \delta, \gamma), \tag{4}$$

## B    $\gamma$-Graph Dimension and ERM Learnability

In this section, we show that $\gamma$-graph dimension determines the learnability of $\mathcal{H} \subseteq [0, 1]^\mathcal{X}$ using *any* ERM learner. We first revisit the notion of partial concept classes which will be useful for deriving our algorithms.

## B.1 Partial Concept Classes and A Naive Approach that Fails

[AHHM22] proposed an extension of the binary PAC model to handle *partial concept classes*, where $\mathcal{H} \subseteq \{0, 1, \star\}^{\mathcal{X}}$, for some input domain $\mathcal{X}$, where $h(x) = \star$ should be thought of as $h$ not knowing the label of $x \in \mathcal{X}$. The main motivation behind their work is that partial classes allow one to conveniently express *data-dependent* assumptions. As an intuitive example, a halfspace with margin is a partial function that is undefined inside the forbidden margin and is a well-defined halfspace outside the margin boundaries. Instead of dealing with concept classes $\mathcal{H} \subseteq \mathcal{Y}^{\mathcal{X}}$ where each concept $h \in \mathcal{H}$ is a **total function** $h : \mathcal{X} \to \mathcal{Y}$, we study **partial concept classes** $\mathcal{H} \subseteq (\mathcal{Y} \cup \{\star\})^{\mathcal{X}}$, where each concept $h$ is now a **partial function** and $h(x) = \star$ means that the function $h$ is **undefined** at $x$. We define the support of $h$ as the set $\mathrm{supp}(h) = \{x \in \mathcal{X} : h(x) \neq \star\}$. Similarly as in the case of total classes, we say that a finite sequence $S = (x_1, y_1, \ldots, x_n, y_n)$ is realizable with respect to $\mathcal{H}$ if there exists some $h^* \in \mathcal{H}$ such that $h^*(x_i) = y_i, \forall i \in [n]$.

An important notion related to partial concept classes is that of *disambiguation*.

**Definition 14** (Disambiguation of Partial Concept Class [AHHM22])**.** *Let $\mathcal{X}$ be an input domain. A total concept class $\overline{\mathcal{H}} \subseteq \{0, 1\}^{\mathcal{X}}$ is a special type of a partial concept. Given some partial concept class $\mathcal{H} \subseteq \{0, 1, \star\}^{\mathcal{X}}$ we say that $\overline{H}$ is a disambiguation of $\mathcal{H}$ if for any finite sequence $S \in (\mathcal{X} \times \{0, 1\})^*$ if $S$ is realizable with respect to $\mathcal{H}$, then $S$ is realizable with respect to $\overline{\mathcal{H}}$.*

Intuitively, by disambiguating a partial concept class we convert it to a total concept class without reducing its "expressivity".

Let us first describe an approach to prove the upper bound, i.e., that if the scaled-graph dimension is finite for all scales then the class is ERM learnable, that does not work. We could perform the following transformation, inspired by the multiclass setting [DSS14]: for any $h \in \mathcal{H} \subseteq [0, 1]^{\mathcal{X}}$, let us consider the function $\widetilde{h} : \mathcal{X} \times [0, 1] \to \{0, 1\}$ with $\widetilde{h}(x, y) = 1$ if and only if $h(x) = y$, $\widetilde{h}(x, y) = 0$ if and only if $\ell(h(x), y) > \varepsilon$ and $\widetilde{h}(x, y) = \star$ otherwise. This induces a new binary *partial* hypothesis class $\widetilde{\mathcal{H}} = \{\widetilde{h} : h \in \mathcal{H}\} \subseteq \{0, 1, \star\}^{\mathcal{X}}$. We note that $\mathbb{D}_{\varepsilon}^{\mathrm{G}}(\mathcal{H}) = \mathrm{VC}(\widetilde{\mathcal{H}})$. However, we cannot use ERM for the partial concept class since in general this approach fails. In particular, a sufficient condition for applying ERM is that $\mathrm{VC}(\{\mathrm{supp}(\widetilde{h}) : \widetilde{h} \in \widetilde{\mathcal{H}}\}) < \infty$.

**Remark 1.** *Predicting $\star$ in [AHHM22] implies a mistake for the setting of partial concept classes. However, in our regression setting, $\star$ is interpreted differently and corresponds to loss at most $\gamma$ which is desirable. In particular, the hard instance for proper learners in the partial concepts paper (see Proposition 4 in [AHHM22]) is good in settings where predicting $\star$ does not count as a mistake, as in our regression case.*

## B.2 Main Result

We are now ready to state the main result of this section. We will prove the next statement.

**Theorem 1.** *Let $\ell$ be the absolute loss function. For every class $\mathcal{H} \subseteq [0, 1]^{\mathcal{X}}$ and for any $\varepsilon, \delta, \gamma \in (0, 1)^3$, the sample complexity bound for realizable PAC regression by any ERM satisfies*

$$\Omega \left( \frac{\mathbb{D}_{\gamma}^{\mathrm{G}}(\mathcal{H}) + \log(1/\delta)}{\varepsilon} \right) \leq \mathcal{M}_{\mathrm{ERM}}(\mathcal{H}; \varepsilon, \delta, \gamma) \leq O \left( \frac{\mathbb{D}_{\gamma}^{\mathrm{G}}(\mathcal{H}) \log(1/\varepsilon) + \log(1/\delta)}{\varepsilon} \right).$$

*In particular, any ERM algorithm $\mathbb{A}$ achieves*

$$\mathop{\mathbf{E}}_{x \sim \mathcal{D}_{\mathcal{X}}} [\ell(A(S; x), h^{\star}(x)] \leq \inf_{\gamma \in [0, 1]} \gamma + \widetilde{\Theta} \left( \frac{\mathbb{D}_{\gamma}^{\mathrm{G}}(\mathcal{H}) + \log(1/\delta)}{n} \right),$$

*with probability at least $1 - \delta$ over $S$ of size $n$.*

*Proof.* We prove the upper bound and the lower bound of the statement separately.

**Upper Bound for the ERM learner.** We deal with the cut-off loss problem with parameters $(\varepsilon, \delta, \gamma) \in (0, 1)^3$. Our proof is based on a technique that uses a "ghost" sample to establish generalization guarantees of the algorithm. Let us denote

$$\mathrm{er}_{\mathcal{D}, \gamma}(h) \triangleq \mathop{\mathbf{Pr}}_{x \sim \mathcal{D}_{\mathcal{X}}} [\ell(h(x), h^{\star}(x)) > \gamma], \tag{1}$$

and for a dataset $z \in (\mathcal{X} \times [0, 1])^n$,

$$\widehat{\mathrm{er}}_{z,\gamma}(h) \triangleq \frac{1}{|z|} \sum_{(x,y) \in z} \mathbb{I}\{\ell(h(x), y) > \gamma\}. \tag{2}$$

We will start by showing the next symmetrization lemma in our setting. Essentially, it bounds the probability that there exists a bad ERM learner by the probability that there exists an ERM learner on a sample $r$ whose performance on a hidden sample $s$ is bad. For a similar result, see Lemma 4.4 in [AB99].

**Lemma 2** (Symmetrization)**.** *Let* $\varepsilon, \gamma \in (0, 1)^2, n > 0$. *Fix* $Z = \mathcal{X} \times [0, 1]$. *Let*

$$Q_{\varepsilon,\gamma} = \{z \in Z^n : \exists h \in \mathcal{H} : \widehat{\mathrm{er}}_{z,\gamma}(h) = 0, \ \mathrm{er}_{\mathcal{D},\gamma}(h) > \varepsilon\} \tag{3}$$

*and*

$$R_{\varepsilon,\gamma} = \{(r, s) \in Z^n \times Z^n : \exists h \in \mathcal{H} : \mathrm{er}_{\mathcal{D},\gamma}(h) > \varepsilon, \ \widehat{\mathrm{er}}_{r,\gamma}(h) = 0, \ \widehat{\mathrm{er}}_{s,\gamma}(h) \geq \varepsilon/2\}. \tag{4}$$

*Then, for* $n \geq c/\varepsilon$*, where* $c$ *is some absolute constant, we have that*

$$\mathcal{D}^n(Q_{\varepsilon,\gamma}) \leq 2\mathcal{D}^{2n}(R_{\varepsilon,\gamma}).$$

*Proof.* We will show that $\mathcal{D}^{2n}(R_{\varepsilon,\gamma}) \geq \frac{\mathcal{D}^n(Q_{\varepsilon,\gamma})}{2}$. By the definition of $R_{\varepsilon,\gamma}$ we can write

$$\mathcal{D}^{2n}(R_{\varepsilon,\gamma}) = \int_{Q_{\varepsilon,\gamma}} \mathcal{D}^n(s : \exists h \in \mathcal{H}, \mathrm{er}_{\mathcal{D},\gamma}(h) > \varepsilon, \widehat{\mathrm{er}}_{r,\gamma}(h) = 0, \widehat{\mathrm{er}}_{s,\gamma}(h) \geq \varepsilon/2) d\mathcal{D}^n(r).$$

For $r \in Q_{\varepsilon,\gamma}$, fix $h_r \in \mathcal{H}$ that satisfies $\widehat{\mathrm{er}}_{r,\gamma}(h_r) = 0, \ \mathrm{er}_{\mathcal{D},\gamma}(h) > \varepsilon$. It suffices to show that for $h_r$

$$\mathcal{D}^n(s : \widehat{\mathrm{er}}_{s,\gamma}(h_r) \geq \varepsilon/2) \geq 1/2.$$

Then, the proof of the lemma follows immediately. Since $\mathrm{er}_{\mathcal{D},\gamma}(h_r) > \varepsilon$, we know that $n \cdot \widehat{\mathrm{er}}_{s,\gamma}(h)$ follows a binomial distribution with probability of success on every try at least $\varepsilon$ and $n$ number of tries. Thus, the multiplicative version of Chernoff's bound gives us

$$\mathcal{D}^n(s : \widehat{\mathrm{er}}_{s,\gamma}(h_r) < \varepsilon/2) \leq e^{-\frac{n \cdot \varepsilon}{8}}.$$

Thus, if $n = c/\varepsilon$, for some appropriate absolute constant $c$ we see that

$$\mathcal{D}^n(s : \widehat{\mathrm{er}}_{s,\gamma}(h_r) < \varepsilon/2) < 1/2,$$

which concludes the proof. $\qquad\square$

Next, we can use a random swap argument to upper bound $\mathcal{D}^{2n}(R_{\varepsilon,\gamma})$ with a quantity that involves a set of permutations over the sample of length $2n$. The main idea behind the proof is to try to leverage the fact that each of the labeled examples is as likely to occur among the first $n$ examples or the last $n$ examples.

Following [AB99], we denote by $\Gamma_n$ the set of all permutations on $\{1, \ldots, 2n\}$ that swap $i$ and $n + i$, for all $i$ that belongs to $\{1, \ldots, n\}$. In other words, for all $\sigma \in \Gamma_n$ and $i \in \{1, \ldots, n\}$ either $\sigma(i) = i, \sigma(n + i) = n + i$ or $\sigma(i) = n + i, \sigma(n + i) = i$. Thus, we can think of $\sigma$ as acting on coordinates where it (potentially) swaps one element from the first half of the sample with the corresponding element on the second half of the sample. For some $z \in Z^{2n}$ we overload the notation and denote $\sigma(z)$ the effect of applying $\sigma$ to the sample $z$.

We are now ready to state the bound. Importantly, it shows that by (uniformly) randomly choosing a permutation $\sigma \in \Gamma_n$ we can bound the probability that a sample falls into the bad set $R_{\varepsilon,\gamma}$ by a quantity that does not depend on the distribution $\mathcal{D}$.

**Lemma 3** (Random Swaps; Adaptation of Lemma 4.5 in [AB99])**.** *Fix* $Z = \mathcal{X} \times [0, 1]$. *Let* $R_{\varepsilon,\gamma}$ *be any subset of* $Z^{2n}$ *and* $\mathcal{D}$ *any probability distribution on* $Z$. *Then*

$$\mathcal{D}^{2n}(R_{\varepsilon,\gamma}) = \mathop{\mathbf{E}}_{z \sim \mathcal{D}^{2n}} \mathop{\mathbf{Pr}}_{\sigma \sim \mathbb{U}(\Gamma_n)} [\sigma(z) \in R_{\varepsilon,\gamma}] \leq \max_{z \in Z^{2n}} \mathop{\mathbf{Pr}}_{\sigma \sim \mathbb{U}(\Gamma_n)} [\sigma(z) \in R_{\varepsilon,\gamma}],$$

*where* $\mathbb{U}(\Gamma_n)$ *is the uniform distribution over the set of swapping permutations* $\Gamma_n$.

*Proof.* First, notice that the bound

$$\mathop{\mathbf{E}}_{z \sim \mathcal{D}^{2n}} \mathop{\mathbf{Pr}}_{\sigma \sim \mathbb{U}(\Gamma_n)}[\sigma(z) \in R_{\varepsilon,\gamma}] \le \max_{z \in Z^{2n}} \mathop{\mathbf{Pr}}_{\sigma \sim \mathbb{U}(\Gamma_n)}[\sigma(z) \in R_{\varepsilon,\gamma}],$$

follows trivially and the maximum exists since $\mathbf{Pr}_{\sigma \sim \mathbb{U}(\Gamma_n)}[\sigma(z) \in R_{\varepsilon,\gamma}]$ takes finitely many values for any finite $n$ and all $z \in Z^{2n}$. Thus, the bulk of the proof is to show that

$$\mathcal{D}^{2n}(R_{\varepsilon,\gamma}) = \mathop{\mathbf{E}}_{z \sim \mathcal{D}^{2n}} \mathop{\mathbf{Pr}}_{\sigma \sim \mathbb{U}(\Gamma_n)}[\sigma(z) \in R_{\varepsilon,\gamma}].$$

First, notice that since example is drawn i.i.d., for any swapping permutation $\sigma \in \Gamma_n$ we have that

$$\mathcal{D}^{2n}(R_{\varepsilon,\gamma}) = \mathcal{D}^{2n}\left(\left\{z \in Z^{2n} : \sigma(z) \in R_{\varepsilon,\gamma}\right\}\right). \tag{5}$$

Thus, the following holds

$$
\begin{aligned}
\mathcal{D}^{2n}(R_{\varepsilon,\gamma}) &= \int_{Z^{2n}} \mathbb{I}\{z \in R_{\varepsilon,\gamma}\} \, d\mathcal{D}^{2n}(z) \\
&= \frac{1}{|\Gamma_n|} \sum_{\sigma \in \Gamma_n} \int_{Z^{2n}} \mathbb{I}\{\sigma(z) \in R_{\varepsilon,\gamma}\} \, d\mathcal{D}^{2n}(z) \\
&= \int_{Z^{2n}} \left( \frac{1}{|\Gamma_n|} \sum_{\sigma \in \Gamma_n} \mathbb{I}\{\sigma(z) \in R_{\varepsilon,\gamma}\} \right) d\mathcal{D}^{2n}(z) \\
&= \mathop{\mathbf{E}}_{z \sim \mathcal{D}^{2n}} \mathop{\mathbf{Pr}}_{\sigma \sim \mathbb{U}(\Gamma_n)}[\sigma(z) \in R_{\varepsilon,\gamma}],
\end{aligned}
$$

where the first equation follows by definition, the second by Equation (5), the third because the number of terms in the summation is finite, and the last one by definition. $\qquad\square$

As a last step we can bound the above RHS by using all possible patterns when $\mathcal{H}$ is (roughly speaking) projected in the sample $rs \in Z^{2n}$.

**Lemma 4** (Bounding the Bad Event). *Fix $Z = \mathcal{X} \times [0,1]$. Let $R_{\varepsilon,\gamma} \subseteq Z^{2n}$ be the set*

$$R_{\varepsilon,\gamma} = \left\{(r,s) \in Z^n \times Z^n : \exists h \in \mathcal{H} : \widehat{\mathrm{er}}_{r,\gamma}(h) = 0, \ \widehat{\mathrm{er}}_{s,\gamma}(h) \ge \varepsilon/2\right\}.$$

*Then*

$$\mathop{\mathbf{Pr}}_{\sigma \sim \mathbb{U}(\Gamma_n)}[\sigma(z) \in R_{\varepsilon,\gamma}] \le (2n)^{O(\mathbb{D}_\gamma^G(\mathcal{H}) \log(2n))} 2^{-n\varepsilon/2}.$$

*Proof.* Throughout the proof we fix $z = (z_1, ..., z_{2n}) \in Z^{2n}$, where $z_i = (x_i, y_i) = (x_i, h^\star(x_i))$ and let $S = \{x_1, ..., x_{2n}\}$. Consider the projection set $\mathcal{H}|_S$. We define a *partial binary* concept class $\mathcal{H}' \subseteq \{0, 1, \star\}^{2n}$ as follows:

$$\mathcal{H}' := \left\{ h' \in \{0,1,\star\}^{2n} : \exists h \in \mathcal{H}|_S : \forall i \in [2n] \begin{cases} h(x_i) = y_i, & h'(i) = 0 \\ \ell(h(x_i), y_i) > \gamma, & h'(i) = 1 \\ 0 < \ell(h(x_i), y_i) \le \gamma, & h'(i) = \star \end{cases} \right\}.$$

Importantly, we note that, by definition, $\mathrm{VC}(\mathcal{H}') \le \mathbb{D}_\gamma^G(\mathcal{H})$.

Currently, we have a partial binary concept class $\mathcal{H}'$. As a next step, we would like to replace the $\star$ symbols and essentially reduce the problem to a total concept class. This procedure is called disambiguation (cf. Definition 14). The next key lemma shows that there exists a compact (in terms of cardinality) disambiguation of a VC partial concept class for finite instance domains.

**Lemma 5** (Compact Disambiguations, see [AHHM22]). *Let $\mathcal{H}$ be a partial concept class on a finite instance domain $\mathcal{X}$ with $\mathrm{VC}(\mathcal{H}) = d$. Then there exists a disambiguation $\overline{\mathcal{H}}$ of $\mathcal{H}$ with size $|\overline{\mathcal{H}}| = |\mathcal{X}|^{O(d \log |\mathcal{X}|)}$.*

This means that there exists a disambiguation $\overline{\mathcal{H}'}$ of $\mathcal{H}'$ of size at most

$$(2n)^{O(\mathbb{D}_\gamma^G(\mathcal{H}) \log(2n))}.$$

Since this (total) binary concept class is finite, we can apply the following union bound argument. We have that $\sigma(z) \in R$ if and only if some $h \in \mathcal{H}$ satisfies

$$\frac{\sum_{i=1}^{n} \mathbb{I}\{\ell(h(x_{\sigma(i)}), y_{\sigma(i)}) > \gamma\}}{n} = 0, \quad \frac{\sum_{i=1}^{n} \mathbb{I}\{\ell(h(x_{\sigma(n+i)}), y_{\sigma(n+i)}) > \gamma\}}{n} \geq \varepsilon/2.$$

We can relate this event with an event about the disambiguated partial concept class $\overline{\mathcal{H}'}$ since the number of 1's can only increase. In particular, for any swapping permutation $\sigma$ of the $2n$ points, if there exists a function in $\mathcal{H}$ that is correct on the first $n$ points and is off by at least $\gamma$ on at least $\varepsilon n/2$ of the remaining $n$ points, then there is a function in the disambiguation $\overline{\mathcal{H}'}$ that is 0 on the first $n$ points and is 1 on those same $\epsilon n/2$ of the remaining points.

If we fix some $\sigma \in \Gamma_n$, and some $\overline{h'} \in \overline{\mathcal{H}'}$ then $\overline{h'}$ is a witness that $\sigma(z) \in R_{\varepsilon,\gamma}$ only if $\forall i \in [n]$ we do not have that $\overline{h'}(i) = 1, \overline{h'}(i+n) = 1$. Thus at least one of $\overline{h'}(i), \overline{h'}(n+i)$ must be zero. Moreover, at least $n\varepsilon/2$ entries must be non-zero. Thus, when we draw random swapping permutation the probability that all the non-zero entries land on the second half of the sample sample is at most $2^{-n\varepsilon/2}$.

Crucially since the number of possible functions is at most $|\overline{\mathcal{H}'}| \leq (2n)^{O(\mathbb{D}_\gamma^G(\mathcal{H}) \log(2n))}$ a union bound gives us that

$$\Pr_{\sigma \sim \mathbb{U}(\Gamma_n)} [\sigma(z) \in R_{\varepsilon,\gamma}] \leq (2n)^{O(\mathbb{D}_\gamma^G(\mathcal{H}) \log(2n))} \cdot 2^{-n\varepsilon/2}.$$

Thus, since $z \in Z^{2n}$ was arbitrary we have that

$$\max_{z \in Z^{2n}} \Pr_{\sigma \mathbb{U}(\Gamma_n)} [\sigma(z) \in R_{\varepsilon,\gamma}] \leq (2n)^{O(\mathbb{D}_\gamma^G(\mathcal{H}) \log(2n))} \cdot 2^{-n\varepsilon/2}.$$

This concludes the proof. $\qquad\square$

**Lower Bound for the ERM learner.** Our next goal is to show that

$$\mathcal{M}_{\mathrm{ERM}}(\mathcal{H}; \varepsilon, \delta, \gamma) \geq C_0 \cdot \frac{\mathbb{D}_\gamma^G(\mathcal{H}) + \log(1/\delta)}{\varepsilon}.$$

To this end, we will show that there exists an ERM learner satisfying this lower bound. This will establish that the finiteness of $\gamma$-graph dimension for any $\gamma \in (0, 1)$, is necessary for PAC learnability using *a worst-case* ERM. It suffices to show that there exists a bad ERM algorithm that requires at least $C_0 \frac{\mathbb{D}_\gamma^G(\mathcal{H}) + \log(1/\delta)}{\varepsilon}$ samples to cut-off PAC learn $\mathcal{H}$. First let us consider the case where $d = \mathbb{D}_\gamma^G(\mathcal{H}) < \infty$ and let $S = \{x_1, ...., x_d\}$ be a $\gamma$-graph-shattered set by $\mathcal{H}$ with witness $f_0$. Consider the ERM learner $\mathcal{A}$ that works as follows: Upon seeing a sample $T \subseteq S$ consistent with $f_0$, $\mathcal{A}$ returns a function $\mathcal{A}(T)$ that is equal to $f_0$ on elements of $T$ and $\gamma$-far from $f_0$ on $S \setminus T$. Such a function exists since $S$ is $\gamma$-graph-shattered with witness $f_0$. Let us take $\delta < 1/100$ and $\varepsilon < 1/12$. Define a distribution over $S \subseteq \mathcal{X}$ such that

$$\mathbf{Pr}[x_1] = 1 - 2\varepsilon, \quad \mathbf{Pr}[x_i] = 2\varepsilon/(d-1), \ \forall i \in \{2, ..., d\}.$$

Let us set $h^\star = f_0$ and consider $m$ samples $\{(z_i, f_0(z_i)\}_{i \in [m]}$. Since we work in the scaled PAC model, $\mathcal{A}$ will make a $\gamma$-error on all examples from $S$ which are not in the sample (since in that case the output will be $\gamma$-far from the true label). Let us take $m \leq \frac{d-1}{6\varepsilon}$. Then, the sample will include at most $(d-1)/2$ examples which are not $x_1$ with probability $1/100$, using Chernoff's bound. Conditioned on that event, this implies that the ERM learner will make a $\gamma$-error with probability at least $\frac{2\varepsilon}{d-1} \cdot (d - 1 - \frac{d-1}{2}) = \varepsilon$, over the random draw of the test point. Thus, $\mathcal{M}_\mathcal{A}(\mathcal{H}; \varepsilon, \delta, \gamma) = \Omega(\frac{d-1}{\varepsilon})$. Moreover, the probability that the sample will only contain $x_1$ is $(1 - 2\varepsilon)^m \geq e^{-4\varepsilon m}$ which is greater that $\delta$ whenever $m \leq \log(1/\delta)/(4\varepsilon)$. This implies that the $\gamma$-cut-off ERM sample complexity is lower bounded by

$$\max \left\{ \frac{d-1}{6\varepsilon}, \frac{\log(1/\delta)}{2\varepsilon} \right\} = C_0 \cdot \frac{\mathbb{D}_\gamma^G(\mathcal{H}) + \log(1/\delta)}{\varepsilon}.$$

Thus $\mathcal{M}_{\mathrm{ERM}}(\mathcal{H}; \varepsilon, \delta, \gamma)$, satisfies the desired bound when the dimension is finite. Finally, it remains to claim about the case where $\mathbb{D}_\gamma^G(\mathcal{H}) = \infty$ for the given $\gamma$. We consider a sequence of $\gamma$-graph-shattered sets $S_n$ with $|S_n| = n$ and repeat the claim for the finite case. This will yield that for any $n$ the cut-off ERM sample complexity is lower bounded by $\Omega((n + \log(1/\delta))/\varepsilon)$ and this yields that $\mathcal{M}_{\mathrm{ERM}}(\mathcal{H}; \varepsilon, \delta, \gamma) = \infty$. $\qquad\square$

However, as Example 4 shows, the optimal learner cannot be proper and as a result, this dimension does not characterize PAC learnability for real-valued regression (there exist classes whose $\gamma$-graph dimension is infinite but are PAC learnable in the realizable regression setting).

## C  $\gamma$-OIG Dimension and Learnability

In this section we identify a dimension characterizing qualitatively and quantitatively what classes of predictors $\mathcal{H} \subseteq [0,1]^{\mathcal{X}}$ are PAC learnable and we provide PAC learners that achieve (almost) optimal sample complexity. In particular, we show the following result.

**Theorem 2.** *Let $\ell$ be the absolute loss function. For every class $\mathcal{H} \subseteq [0,1]^{\mathcal{X}}$ and for any $\varepsilon, \delta, \gamma \in (0,1)^3$, the sample complexity bound for realizable PAC regression satisfies*

$$\Omega\left(\frac{\mathbb{D}_{2\gamma}^{\mathrm{OIG}}(\mathcal{H})}{\varepsilon}\right) \leq \mathcal{M}(\mathcal{H}; \varepsilon, \delta, \gamma) \leq O\left(\frac{\mathbb{D}_{\gamma}^{\mathrm{OIG}}(\mathcal{H})}{\varepsilon} \log^2 \frac{\mathbb{D}_{\gamma}^{\mathrm{OIG}}(\mathcal{H})}{\varepsilon} + \frac{1}{\varepsilon} \log \frac{1}{\delta}\right).$$

*In particular, there exists an algorithm $A$ such that*

$$\mathop{\mathbf{E}}_{x \sim \mathcal{D}_{\mathcal{X}}}[\ell(A(S; x), h^{\star}(x)] \leq \inf_{\gamma \in [0,1]} \gamma + \widetilde{\Theta}\left(\frac{\mathbb{D}_{\gamma}^{\mathrm{OIG}}(\mathcal{H}) + \log(1/\delta)}{n}\right),$$

*with probability at least $1 - \delta$ over $S \sim \mathcal{D}^n$.*

**A finite fat-shattering dimension implies a finite OIG dimension.**  Let $\mathcal{F} \subseteq [0,1]^{\mathcal{X}}$ be a function class with finite $\gamma$-fat shattering dimension for any $\gamma > 0$. We show that $\mathbb{D}_{\gamma}^{\mathrm{OIG}}(\mathcal{F})$ is upper bounded (up to constants and log factors) by $\mathbb{D}_{c\gamma}^{\mathrm{fat}}(\mathcal{F})$, for some $c > 0$, where the OIG dimension is defined with respect to the $\ell_1$ loss. Note that the opposite direction does not hold. Example 1 exhibits a function class with an infinite fat-shattering dimension that can be learned with a single example, and as a result, the OIG dimension has to be finite. On the one hand, we have an upper bound on the sample complexity of $O\left(\frac{1}{\epsilon}\left(\mathbb{D}_{\tilde{c}\varepsilon}^{\mathrm{fat}}(\mathcal{F}) \log^2 \frac{1}{\epsilon} + \log \frac{1}{\delta}\right)\right)$, for any $\varepsilon, \delta \in (0,1)$. See sections 19.6 and 20.4 about the restricted model in [AB99]. On the other hand, we prove in Lemma 6 a lower bound on the sample complexity of $\Omega\left(\frac{\mathbb{D}_{2\varepsilon}^{\mathrm{OIG}}(\mathcal{F})}{\epsilon}\right)$, for any $\varepsilon \in (0,1)$, and so $\mathbb{D}_{\gamma}^{\mathrm{OIG}}(\mathcal{F})$ is upper bounded by $\mathbb{D}_{c\gamma}^{\mathrm{fat}}(\mathcal{F})$ up to constants and log factors.

### C.1  Proof of the Lower Bound

**Lemma 6.** *[Lower Bound of PAC Regression] Let $A$ be any learning algorithm and $\varepsilon, \delta, \gamma \in (0,1)^3$ such that $\delta < \varepsilon$. Then,*

$$\mathcal{M}_A(\mathcal{H}; \varepsilon, \delta, \gamma) \geq \Omega\left(\frac{\mathbb{D}_{2\gamma}^{\mathrm{OIG}}(\mathcal{H})}{\varepsilon}\right).$$

*Proof.* Let $n_0 = \mathbb{D}_{2\gamma}^{\mathrm{OIG}}(\mathcal{H})$. Let $n \in \mathbb{N}, 1 < n \leq n_0$. We know that for each such $n$ there exists some $S \in \mathcal{X}^n$ such that the one-inclusion graph of $\mathcal{H}|_S$ has the property that: there exists a finite subgraph $G = (V, E)$ of $G_{\mathcal{H}|_S}^{\mathrm{OIG}}$ such that for any orientation $\sigma : E \to V$ of the subgraph, there exists a vertex $v \in V$ with $\mathrm{outdeg}(v; \sigma, 2\gamma) > \mathbb{D}_{2\gamma}^{\mathrm{OIG}}(\mathcal{H})/3$.

Given the learning algorithm $A : (\mathcal{X} \times [0,1])^{\star} \times \mathcal{X} \to [0,1]$, we can describe an orientation $\sigma_A$ of the edges in $E$. For any vertex $v = (v_1, \ldots, v_n) \in V$ let $P_v$ be the distribution over $(x_1, v_1), \ldots, (x_n, v_n)$ defined as

$$P_v((x_1, v_1)) = 1 - \varepsilon, \quad P_v((x_t, v_t)) = \frac{\varepsilon}{n-1}, \quad t \in \{2, \ldots, n\}.$$

Let $m = n/(2\varepsilon)$. For each vertex $v \in V$ and direction $t \in [n]$, consider the hyperedge $e_{t,v}$. For each $u \in e_{t,v}$ we define

$$p_t(u) = \mathop{\mathbf{Pr}}_{S \sim P_u^m}[\ell(A(S; x_t), u_t) > \gamma | (x_t, u_t) \notin S],$$

and let $C_{e_{t,v}} = \{u \in e_{t,v} : p_t(u) < 1/2\}$. If $C_{e_{t,v}} = \emptyset$, we orient the edge $e_{t,v}$ arbitrarily. Since for all $u, v \in e_{t,v}$ the distributions $P_u^m, P_v^m$ conditioned on the event that $(x_t, u_t), (x_t, v_t)$ respectively

are not in $S$ are the same, we can see that $\forall u, v \in C_{e_{t,v}}$ it holds that $\ell(u_t, v_t) \leq 2\gamma$. We orient the edge $e_{t,v}$ using an arbitrary element of $C_{e_{t,v}}$.

Because of the previous discussion, we can bound from above the out-degree of all vertices $v \in V$ with respect to the orientation $\sigma_A$ as follows:

$$\text{outdeg}(v; \sigma_A, 2\gamma) \leq \sum_t \mathbb{I}\{p_t(v) \geq 1/2\} \leq 1 + 2\sum_{t=2}^n \Pr_{S \sim P_v^m}[\ell(A(S, x_t), y_t) > \gamma | (x_t, y_t) \notin S].$$

Notice that

$$\Pr_{S \sim P_v^m}[\ell(A(S, x_t), y_t) > \gamma | (x_t, y_t) \notin S] = \frac{\mathbf{Pr}_{S \sim P_v^m}[\{\ell(A(S, x_t), y_t) > \gamma\} \wedge \{(x_t, y_t) \notin S\}]}{\mathbf{Pr}_{S \sim P_v^m}[(x_t, y_t) \notin S]},$$

and by the definition of $P_v$, we have that

$$\Pr_{S \sim P_v^m}[(x_t, y_t) \notin S] = \left(1 - \frac{\varepsilon}{n-1}\right)^m \geq 1 - \frac{n}{2(n-1)},$$

since $m = n/(2\varepsilon)$. Combining the above, we get that

$$\text{outdeg}(v; \sigma_A, 2\gamma) \leq 1 + 2\left(1 - \frac{n}{2(n-1)}\right)\sum_{t=2}^n \mathop{\mathbf{E}}_{S \sim P_v^m}[\mathbb{I}\{\ell(A(S, x_t), y_t) > \gamma\} \cdot \mathbb{I}\{(x_t, y_t) \notin S\}],$$

and so

$$
\begin{aligned}
\text{outdeg}(v; \sigma_A, 2\gamma) &\leq 1 + 2\left(1 - \frac{n}{2(n-1)}\right)\mathop{\mathbf{E}}_{S \sim P_v^m}\left[\sum_{t=2}^n \mathbb{I}\{\ell(A(S, x_t), y_t) > \gamma\}\right] \\
&= 1 + 2\left(1 - \frac{n}{2(n-1)}\right)\frac{n-1}{\varepsilon}\mathop{\mathbf{E}}_{S \sim P_v^m}\left[\frac{\varepsilon}{n-1}\sum_{t=2}^n \mathbb{I}\{\ell(A(S, x_t), y_t) > \gamma\}\right] \\
&\leq 1 + 2\left(1 - \frac{n}{2(n-1)}\right)\frac{n-1}{\varepsilon}\mathop{\mathbf{E}}_{S \sim P_v^m}\left[\mathop{\mathbf{E}}_{(x,y)\sim P_v}[\mathbb{I}\{\ell(A(S; x), y) > \gamma\}]\right] \\
&= 1 + 2\left(1 - \frac{n}{2(n-1)}\right)\frac{n-1}{\varepsilon}\mathop{\mathbf{E}}_{S \sim P_v^m}\left[\Pr_{(x,y)\sim P_v}[\ell(A(S; x), y) > \gamma]\right] \\
&\leq 1 + \frac{n-2}{\varepsilon}\mathop{\mathbf{E}}_{S \sim P_v^m}\left[\Pr_{(x,y)\sim P_v}[\ell(A(S; x), y) > \gamma]\right]
\end{aligned}
$$

By picking "hard" distribution $\mathcal{D} = P_{v^\star}$, where $v^\star \in \arg\max_{v' \in V} \text{outdeg}(v'; \sigma_A, 2\gamma)$ we get that that

$$
\begin{aligned}
\mathop{\mathbf{E}}_{S \sim P_{v^\star}^m}\left[\Pr_{(x,y)\sim P_{v^\star}}[\ell(A(S; x), y) > \gamma]\right] &\geq (\text{outdeg}(v^\star; \sigma_A, 2\gamma) - 1) \cdot \frac{\varepsilon}{n-2} \\
&\geq \frac{\varepsilon}{6},
\end{aligned}
$$

since $\text{outdeg}(v^\star; \sigma_A, 2\gamma) > n/3$. By picking $n = n_0$ we see that when the learner uses $m = n_0/\varepsilon$ samples then its expected error is at least $\varepsilon/6$. Notice that when the learner uses $m' = \mathcal{M}_A(\mathcal{H}; \varepsilon, \delta, \gamma)$ samples we have that

$$\mathop{\mathbf{E}}_{S \sim P_{v^\star}^{m'}}\left[\Pr_{(x,y)\sim P_{v^\star}}[\ell(A(S; x), y) > \gamma]\right] \leq \delta + (1-\delta)\varepsilon \leq \delta + \varepsilon \leq 2\varepsilon.$$

Thus, we see that for any algorithm $A$

$$\mathcal{M}_A(\mathcal{H}; \varepsilon, \delta, \gamma) \geq \Omega\left(\frac{\mathbb{D}_{2\gamma}^{\text{OIG}}(\mathcal{H})}{\varepsilon}\right),$$

hence

$$\mathcal{M}(\mathcal{H}; \varepsilon, \delta, \gamma) \geq \Omega\left(\frac{\mathbb{D}_{2\gamma}^{\text{OIG}}(\mathcal{H})}{\varepsilon}\right).$$

$\square$

## C.2 Proof of the Upper Bound

Let us present the upper bound. For this proof, we need three tools: we will provide a weak learner based on the scaled one-inclusion graph, a boosting algorithm for real-valued functions, and consistent sample compression schemes for real-valued functions.

To this end, we introduce the one-inclusion graph (OIG) algorithm $\mathbb{A}_\gamma^{\mathrm{OIG}}$ for realizable regression at scale $\gamma$.

### C.2.1 Scaled One-Inclusion Graph Algorithm and Weak Learning

First, we show that every scaled orientation $\sigma$ of the one-inclusion graph gives rise to a learner $\mathbb{A}_\sigma$ whose expected absolute loss is upper bounded by the maximum out-degree induced by $\sigma$.

**Lemma 7** (From Orientations to Learners). *Let $\mathcal{D}_\mathcal{X}$ be a distribution over $\mathcal{X}$ and $h^\star \in \mathcal{H} \subseteq [0,1]^\mathcal{X}$, let $n \in \mathbb{N}$ and $\gamma \in (0,1)$. Then, for any orientation $\sigma : E_n \to V_n$ of the scaled-one-inclusion graph $G_\mathcal{H}^{\mathrm{OIG}} = (V_n, E_n)$, there exists a learner $\mathbb{A}_\sigma : (\mathcal{X} \times [0,1])^{n-1} \to [0,1]^\mathcal{X}$, such that*

$$\mathop{\mathbf{E}}_{S \sim \mathcal{D}_\mathcal{X}^{n-1}} \left[ \mathop{\mathbf{Pr}}_{x \sim \mathcal{D}_\mathcal{X}} [\ell(\mathbb{A}_\sigma(x), h^\star(x)) > \gamma] \right] \le \frac{\max_{v \in V_n} \mathrm{outdeg}(v; \sigma, \gamma)}{n},$$

*where $\mathbb{A}_\sigma$ is trained using a sample $S$ of size $n-1$ realized by $h^\star$.*

---

**Algorithm 1** From orientation $\sigma$ to learner $\mathbb{A}_\sigma$

**Input:** An $\mathcal{H}$-realizable sample $\{(x_i, y_i)\}_{i=1}^{n-1}$ and a test point $x \in \mathcal{X}$, $\gamma \in (0,1)$.
**Output:** A prediction $\mathbb{A}_\sigma(x)$.

1. Create the one-inclusion graph $G_{\mathcal{H}|_{(x_1,\ldots,x_{n-1},x)}}^{\mathrm{OIG}}$.

2. Consider the edge in direction $n$ defined by the realizable sample $\{(x_i, y_i)\}_{i=1}^{n-1}$; let

$$e = \{h \in \mathcal{H}|_{(x_1,\ldots,x_{n-1},x)} : \forall i \in [n-1] \; h(i) = y_i\}.$$

3. Return $\mathbb{A}_\sigma(x) = \sigma(e)(n)$.

---

*Proof.* By the classical leave-one-out argument, we have that

$$\mathop{\mathbf{E}}_{S \sim \mathcal{D}_\mathcal{X}^{n-1}} \left[ \mathop{\mathbf{Pr}}_{x \sim \mathcal{D}_\mathcal{X}} [\ell(\mathbb{A}_\sigma(x), h^\star(x)) > \gamma] \right] = \mathop{\mathbf{E}}_{(S,(x,y)) \sim \mathcal{D}^n} [\mathbb{I}\{\ell(h_S(x), y) > \gamma\}] = \mathop{\mathbf{E}}_{S' \sim \mathcal{D}^n, I \sim \mathbb{U}([n])} [\mathbb{I}\{\ell(h_{S'_{-I}}(x'_I), y'_I) > \gamma\}],$$

where $h_S$ is the predictor $\mathbb{A}_\sigma$ using the examples $S$, and $\mathbb{U}([n])$ is the uniform distribution on $\{1, \ldots, n\}$. Now for every fixed $S'$ we have that

$$\mathop{\mathbf{E}}_{I \sim \mathbb{U}([n])} [\mathbb{I}\{\ell(h_{S'_{-I}}(x'_I), y'_I) > \gamma\}] = \frac{1}{n} \sum_{i \in [n]} \mathbb{I}\{\ell(\sigma(e_i)(i), y'_i) > \gamma\} = \frac{\mathrm{outdeg}(y'; \sigma, \gamma)}{n},$$

where $y'$ is the node of the scaled OIG that corresponds to the true labeling of $S'$. By taking expectation over $S' \sim \mathcal{D}^n$ we get that

$$\mathop{\mathbf{E}}_{S' \sim \mathcal{D}^n, I \sim \mathbb{U}([n])} [\mathbb{I}\{\ell(h_{S'_{-I}}(x'_I), y'_I) > \gamma\}] \le \mathop{\mathbf{E}}_{S' \sim \mathcal{D}^n} \left[ \frac{\mathrm{outdeg}(y'; \sigma, \gamma)}{n} \right] \le \frac{\max_{v \in V_n} \mathrm{outdeg}(v; \sigma, \gamma)}{n}.$$

$\square$

Equipped with the previous result, we are now ready to show that when the learner gets at least $\mathbb{D}_\gamma^{\mathrm{OIG}}(\mathcal{H})$ samples as its training set, then its expected $\gamma$-cutoff loss is bounded away from $1/2$.

**Lemma 8** (Scaled OIG Guarantee (Weak Learner)). *Let $\mathcal{D}_\mathcal{X}$ be a distribution over $\mathcal{X}$ and $h^\star \in \mathcal{H} \subseteq [0,1]^\mathcal{X}$, and $\gamma \in (0,1)$. Then, for all $n > \mathbb{D}_\gamma^{\mathrm{OIG}}(\mathcal{H})$ there exists an orientation $\sigma^\star$ such that for the prediction error of the one-inclusion graph algorithm $\mathbb{A}_{\sigma^\star}^{\mathrm{OIG}} : (\mathcal{X} \times [0,1])^{n-1} \times \mathcal{X} \to [0,1]$, it holds that*

$$\mathop{\mathbf{E}}_{S \sim \mathcal{D}_\mathcal{X}^{n-1}} \left[ \mathop{\mathbf{Pr}}_{x \sim \mathcal{D}_\mathcal{X}} [\ell(\mathbb{A}_{\sigma^\star}^{\mathrm{OIG}}(x), h^\star(x)) > \gamma] \right] \le 1/3.$$

*Proof.* Fix $\gamma \in (0,1)$. Assume that $n > \mathbb{D}_\gamma^{\text{OIG}}(\mathcal{H})$ and let $G_{\mathcal{H}|_{(S,x)}}^{\text{OIG}} = (V_n, E_n)$ be the possibly infinite scaled one-inclusion graph. By the definition of the $\gamma$-OIG dimension (see Definition 9), for every finite subgraph $G = (V, E)$ of $G_{\mathcal{H}|_{(S,x)}}^{\text{OIG}}$ there exists an orientation $\sigma : E \to V$ such that for every vertex in $G$ the out-degree is at most $n/3$, i.e.,

$$\forall S \in \mathcal{X}^n, \, \forall \text{ finite } G = (V,E) \text{ of } G_{\mathcal{H}|_S}^{\text{OIG}}, \, \exists \text{ orientation } \sigma_E \text{ s.t. } \forall v \in V, \text{ it holds } \operatorname{outdeg}(v; \sigma_E, \gamma) \le n/3 \, .$$

First, we need to create an orientation of the whole (potentially infinite) one-inclusion graph.

We will create this orientation using the compactness theorem of first-order logic which states that a set of formulas $\Phi$ is satisfiable if and only if it is finitely satisfiable, i.e., every finite subset $\Phi' \subseteq \Phi$ is satisfiable. Let $G_{\mathcal{H}|_S}^{\text{OIG}} = (V_n, E_n)$ be the (potentially infinite) one-inclusion graph of $\mathcal{H}|_S$. Let $\mathcal{Z}$ be the set of pairs $z = (v, e) \in V_n \times E_n$ so that $v \in e$. Our goal is to assign binary values to each $z \in \mathcal{Z}$. We define the following sets of formulas:

- For each $e \in E_n$ we let $\Phi_e := \exists$ exactly one $v \in e : z(v, e) = 1$.

- For each $v \in V_n$ we let $\Phi_v := \exists$ at most $n/3$ different $e_{i,f} \in E_n : v \in e_{i,f} \wedge (\exists v' \in e_{i,f} : (z(v', e) = 1 \wedge \ell(v_i', v_i) > \gamma))$

It is not hard to see that each $\Phi_e, \Phi_v$ can be expressed in first-order logic. Then, we define

$$\Phi := \left( \bigcap_{e \in E_n} \Phi_e \right) \cap \left( \bigcap_{v \in V_n} \Phi_v \right) .$$

Notice that an orientation of the edges of $G_{\mathcal{H}|_S}^{\text{OIG}}$ is equivalent to picking an assignment of the elements of $\mathcal{Z}$ that satisfies all the $\Phi_e$. Moreover, notice that for such an assignment, if all the $\Phi_v$ are satisfied then then maximum $\gamma$-scaled out-degree of $G_{\mathcal{H}|_S}^{\text{OIG}}$ is at most $n/3$.

We will now show that $\Phi$ is finitely satisfiable. Let $\Phi'$ be a finite subset of $\Phi$ and let $E' \subseteq E_n, V' \subseteq V_n$, be the set of edges, vertices that appear in $\Phi'$, respectively. If $V' = \emptyset$, then we can orient the edges in $E'$ arbitrarily and satisfy $\Phi'$. Similarly, if $E' = \emptyset$ we can let all the $z(e, v) = 0$ and satisfy all the $\Phi_v, v \in V'$. Thus, assume that both sets are non-empty. Consider the finite subgraph of $G_{\mathcal{H}|_S}^{\text{OIG}}$ that is induced by $V'$ and let $E''$ be the set of edges of this subgraph. For every edge $e \in E' \setminus E''$[4], pick an arbitrary orientation, i.e, for exactly one $v \in e$ set $z(e, v) = 1$ and for the remaining $v' \in e$ set $z(e, v') = 0$. By the definition of $\mathbb{D}_\gamma^{\text{OIG}}(\mathcal{H})$ there is an orientation $\sigma_{E''}$ of the edges in $E''$ such that $\forall v \in V' \operatorname{outdeg}(v; \sigma_{E''}, \gamma) \le n/3$. For every $e \in E''$ pick the assignment of all the $z(v, e), v \in e$, according to the orientation $\sigma_{E''}$. Thus, because of the maximum out-degree property of $\sigma_{E''}$ we described before, we can also see that all the $\Phi_v, v \in V'$, are satisfied. Hence, we have shown that $\Phi$ is finitely satisfiable, so it is satisfiable. This assignment on $z(v, e)$ induces an orientation $\sigma^\star$ under which all the vertices of the one-inclusion graph have out-degree at most $n/3$.

We will next use the orientation $\sigma^\star$ of $G_{\mathcal{H}|_S}^{\text{OIG}} = (V_n, E_n)$ to design a learner $\mathbb{A}_{\sigma^\star}^{\text{OIG}} : (\mathcal{X} \times [0,1])^{n-1} \times \mathcal{X} \to [0,1]$, invoking Lemma 7. In particular, we get that, from Lemma 7 with the chosen orientation,

$$\mathop{\mathbf{E}}_{S \sim \mathcal{D}_\mathcal{X}^{n-1}} \left[ \mathop{\mathbf{Pr}}_{x \sim \mathcal{D}_\mathcal{X}} \left[ \ell(\mathbb{A}_{\sigma^\star}^{\text{OIG}}(x), h^\star(x)) > \gamma \right] \right] \le \frac{\max_{v \in V_n} \operatorname{outdeg}(v; \sigma^\star, \gamma)}{n} \le 1/3 \, ,$$

which concludes the proof. $\qquad \square$

### C.2.2 Boosting Real-Valued Functions

**Definition 15** (Weak Real-Valued Learner). *Let $\ell$ be a loss function. Let $\gamma \in [0,1]$, $\beta \in (0, \frac{1}{2})$, and $\mathcal{H} \subseteq [0,1]^\mathcal{X}$. For a distribution $\mathcal{D}_\mathcal{X}$ over $\mathcal{X}$ and true target function $h^\star \in \mathcal{H}$, we say that $f : \mathcal{X} \to [0,1]$ is $(\gamma, \beta)$-weak learner with respect to $\mathcal{D}_\mathcal{X}$ and $h^\star$, if*

$$\mathop{\mathbf{Pr}}_{x \sim \mathcal{D}_\mathcal{X}} \left[ \ell(f(x), h^\star(x)) > \gamma \right] < \frac{1}{2} - \beta.$$

---

[4] Since the edges in $E''$ are of finite length, we first need to map them to the appropriate edges in $E'$.

Following [HKS19], we define the weighted median as

$$\text{Median}(y_1, \ldots, y_T; \alpha_1, \ldots, \alpha_T) = \min\left\{ y_j : \frac{\sum_{t=1}^T \alpha_t \mathbb{I}[y_j < y_t]}{\sum_{t=1}^T \alpha_t} < \frac{1}{2} \right\},$$

and the weighted quantiles, for $\theta \in [0, 1/2]$, as

$$Q_\theta^+(y_1, \ldots, y_T; \alpha_1, \ldots, \alpha_T) = \min\left\{ y_j : \frac{\sum_{t=1}^T \alpha_t \mathbb{I}[y_j < y_t]}{\sum_{t=1}^T \alpha_t} < \frac{1}{2} - \theta \right\}$$

$$Q_\theta^-(y_1, \ldots, y_T; \alpha_1, \ldots, \alpha_T) = \max\left\{ y_j : \frac{\sum_{t=1}^T \alpha_t \mathbb{I}[y_j > y_t]}{\sum_{t=1}^T \alpha_t} < \frac{1}{2} - \theta \right\},$$

and we let $Q_\theta^+(x) = Q_\theta^+(h_1(x), \ldots, h_T(x); \alpha_1, \ldots, \alpha_T), Q_\theta^-(x) = Q_\theta^-(h_1(x), \ldots, h_T(x); \alpha_1, \ldots, \alpha_T)$, where $h_1, \ldots, h_T, \alpha_1, \ldots, \alpha_T$ are the values returned by Algorithm 2. The following guarantee holds for this procedure.

**Lemma 9** (MedBoost guarantee [Kég03])**.** *Let $\ell$ be the absolute loss and $S = \{(x_i, y_i)\}_{i=1}^m$, $T = O\big(\frac{1}{\theta^2} \log(m)\big)$. Let $h_1, \ldots, h_T$ and $\alpha_1, \ldots, \alpha_T$ be the functions and coefficients returned from MedBoost. For any $i \in \{1, \ldots, m\}$ it holds that*

$$\max\left\{ \ell\left(Q_{\theta/2}^+(x_i), y_i\right), \ell\left(Q_{\theta/2}^-(x_i), y_i\right) \right\} \leq \gamma.$$

---

**Algorithm 2** MedBoost [Kég03]

---

**Input:** $S = \{(x_i, y_i)\}_{i=1}^m$.
**Parameters:** $\gamma, \beta, T$.
**Initialize** $\mathcal{P}_1 = \text{Uniform}(S)$.
For $t = 1, \ldots, T$:

1. Find a $(\gamma, \beta)$-weak learner $h_t$ with respect to $(x_i, y_i) \sim \mathcal{P}_t$, using a subset $S_t \subseteq S$.

2. For $i = 1, \ldots, m$:

   (a) Set $w_i^{(t)} = 1 - 2\mathbb{I}\big[\ell(h_t(x_i), y_i) > \gamma\big]$.

   (b) Set $\alpha_t = \frac{1}{2} \log\left( \frac{(1-\gamma) \sum_{i=1}^n \mathcal{P}_t(x_i, y_i) \mathbb{I}\big[w_i^{(t)}=1\big]}{(1+\gamma) \sum_{i=1}^n \mathcal{P}_t(x_i, y_i) \mathbb{I}\big[w_i^{(t)}=-1\big]} \right).$

   (c) • If $\alpha_t = \infty$: return $T$ copies of $h_t$, $(\alpha_1 = 1, \ldots, \alpha_T = 1)$, and $S_t$.
       • Else: $P_{t+1}(x_i, y_i) = P_t(x_i, y_i) \exp(-\alpha_t w_i^t) / Z_t$, where $Z_t = \sum_{j=1}^n \mathcal{P}_t(x_j, y_j) \exp(-\alpha_t w_j^t)$.

**Output:** Functions $h_1, \ldots, h_T$, coefficients $\alpha_1, \ldots, \alpha_T$ and sets $S_1, \ldots, S_T$.

---

### C.2.3 Generalization via Sample Compression Schemes

Sample compression scheme is a classic technique for proving generalization bounds, introduced by [LW86, FW95]. These bounds proved to be useful in numerous learning settings, such as binary classification [GHST05, MY16, BHMZ20], multiclass classification [DSBDSS15, DSS14, DMY16, BCD+22], regression [HKS18, HKS19], active learning [WHEY15], density estimation [ABDH+20], adversarially robust learning [MHS19, MHS20, MHS21, MHS22, AHM22, AH22], learning with partial concepts [AHHM22], and showing Bayes-consistency for nearest-neighbor methods [GKN14, KSW17]. As a matter of fact, compressibility and learnability are known to be equivalent for general learning problems [DMY16]. Another remarkable result by [MY16] showed that VC classes enjoy a sample compression that is independent of the sample size.

We start with a formal definition of a sample compression scheme.

**Definition 16** (Sample compression scheme)**.** *A pair of functions $(\kappa, \rho)$ is a sample compression scheme of size $\ell$ for class $\mathcal{H}$ if for any $n \in \mathbb{N}$, $h \in \mathcal{H}$ and sample $S = \{(x_i, h(x_i))\}_{i=1}^n$, it holds for the compression function that $\kappa(S) \subseteq S$ and $|\kappa(S)| \leq \ell$, and the reconstruction function $\rho(\kappa(S)) = \hat{h}$ satisfies $\hat{h}(x_i) = h(x_i)$ for any $i \in [n]$.*

We show a generalization bound that scales with the sample compression size. The proof follows from [LW86].

**Lemma 10** (Sample compression scheme generalization bound). *Fix a margin $\gamma \in [0, 1]$. For any $k \in \mathbb{N}$ and fixed function $\phi : (\mathcal{X} \times [0, 1])^k \to [0, 1]^{\mathcal{X}}$, for any distribution $\mathcal{D}$ over $\mathcal{X} \times [0, 1]$ and any $m \in \mathbb{N}$, for $S = \{(x_i, y_i)\}_{i \in [m]}$ i.i.d. $\mathcal{D}$-distributed random variables, if there exist indices $i_1, ..., i_k \in [m]$ such that $\sum_{(x,y) \in S} \mathbb{I}\{\ell(\phi((x_{i_1}, y_{i_1}), ..., (x_{i_k}, y_{i_k}))(x), y) > \gamma\} = 0$, then*

$$\mathop{\mathbf{E}}_{(x,y) \sim \mathcal{D}} [\mathbb{I}\{\ell(\phi((x_{i_1}, y_{i_1}), ..., (x_{i_k}, y_{i_k}))(x), y) > \gamma\}] \leq \frac{1}{m - k}(k \log m + \log(1/\delta)).$$

*with probability at least $1 - \delta$ over $S$.*

*Proof.* Let us define $\widehat{\ell}_\gamma(h; S) = \frac{1}{|S|} \sum_{(x,y) \in S} \mathbb{I}\{\ell(h(x), y) > \gamma\}$ and $\ell_\gamma(h; \mathcal{D}) = \mathbf{E}_{(x,y) \sim \mathcal{D}} [\mathbb{I}\{\ell(h(x), y) > \gamma\}]$. For any indices $i_1, ..., i_k \in [m]$, the probability of the bad event

$$\mathop{\mathbf{Pr}}_{S \sim \mathcal{D}^m} [\widehat{\ell}_\gamma(\phi((x_{i_1}, y_{i_1}), ..., (x_{i_k}, y_{i_k})); S) = 0 \wedge \ell_\gamma(\phi((x_{i_1}, y_{i_1}), ..., (x_{i_k}, y_{i_k})); \mathcal{D}) > \varepsilon]$$

is at most

$$\mathbf{E}\left[ \mathbb{I}\{\ell_\gamma(\phi(\{(x_{i_j}, y_{i_j})\}_{j \in [k]}); \mathcal{D}) > \varepsilon\} \mathbf{Pr}[\widehat{\ell}_\gamma(\phi(\{(x_{i_j}, y_{i_j})\}_{j \in [k]}); S \setminus \{(x_{i_j}, y_{i_j})\}_{j \in [k]}) = 0 | \{(x_{i_j}, y_{i_j})\}_{j \in [k]}] \right]$$
$$< (1 - \varepsilon)^{m-k}$$

where the expectation is over $(x_{i_1}, y_{i_1}), ..., (x_{i_k}, y_{i_k})$ and the inner probability is over $S \setminus (x_{i_1}, y_{i_1}), ..., (x_{i_k}, y_{i_k})$. Taking a union bound over all $m^k$ possible choices for the $k$ indices, we get that the bad event occurs with probability at most

$$m^k \exp(-\varepsilon(m - k)) \leq \delta \Rightarrow \varepsilon = \frac{1}{m - k}(k \log m + \log(1/\delta)).$$

$\square$

### C.3 Putting it Together

We now have all the necessary ingredients in place to prove the upper bound of Theorem 2. First, we use Lemma 8 on a sample of size $n_0 = \mathbb{D}_\gamma^{\mathrm{OIG}}(\mathcal{H})$ to obtain a learner which makes $\gamma$-errors with probability at most $1/3$[5]. Then, we use the boosting algorithm we described (see Algorithm 2) to obtain a learner that does not make any $\gamma$-mistakes on the training set. Notice that the boosting algorithm on its own does not provide any guarantees about the generalization error of the procedure. This is obtained through the sample compression result we described in Appendix C.2.3. Since we run the boosting algorithm for a few rounds on a sample whose size is small, we can provide a sample compression scheme following the approach of [DMY16, HKS19].

**Lemma 11** (Upper Bound of PAC Regression). *Let $\mathcal{H} \subseteq [0, 1]^{\mathcal{X}}$ and $\varepsilon, \delta, \gamma \in (0, 1)^3$. Then,*

$$\mathcal{M}(\mathcal{H}; \varepsilon, \delta, \gamma) \leq O\left( \frac{\mathbb{D}_\gamma^{\mathrm{OIG}}(\mathcal{H})}{\varepsilon} \log^2 \frac{\mathbb{D}_\gamma^{\mathrm{OIG}}(\mathcal{H})}{\varepsilon} + \frac{1}{\varepsilon} \log \frac{1}{\delta} \right).$$

*Proof.* Let $n$ be the number of samples $S = ((x_1, y_1), ..., (x_n, y_n))$ that are available to the learner, $n_0 = \mathbb{D}_\gamma^{\mathrm{OIG}}(\mathcal{H})$ and let $A$ be the algorithm obtained from Lemma 8. We have that

$$\mathop{\mathbf{E}}_{S \sim \mathcal{D}_{\mathcal{X}}^{n_0-1}} \left[ \mathop{\mathbf{Pr}}_{x \sim \mathcal{D}_{\mathcal{X}}} [\ell(A(S; x), h^\star(x)) > \gamma] \right] \leq 1/3.$$

This means that, for any distribution $\mathcal{D}_{\mathcal{X}}$ and any labeling function $h^\star \in \mathcal{H}$ we can draw a sample $S^\star = ((x_1, y_1), ..., (x_{n_0-1}, y_{n_0-1}))$ with non-zero probability such that

$$\mathop{\mathbf{Pr}}_{x \sim \mathcal{D}_{\mathcal{X}}} [\ell(A(S^\star; x), h^\star(x)) > \gamma] \leq \frac{1}{3}.$$

---

[5]In expectation over the training set.

Notice that such a classifier is a $(\gamma, 1/6)$-weak learner (see Definition 15). Thus, by executing the MedBoost algorithm (see Algorithm 2) for $T = O(\log n)$ rounds we obtain a classifier $\hat{h} : \mathcal{X} \to \mathbb{R}$ such that, $\ell(\hat{h}(x_i), y_i) \leq \gamma, \forall i \in [n]$. We underline that the subset $S_t$ that is used in line 1 of Algorithm 2 has size at most $n_0$, for all rounds $t \in [T]$. Thus, the total number of samples that is used by MedBoost is at most $O(n_0 \log n)$. Hence, following the approach of [MY16] we can encode the classifiers produced by MedBoost as a compression set that consists of $k = O(n_0 \log n)$ samples that were used to train the classifiers along with $k \log k$ extra bits that indicate their order. Thus, using generalization based on sample compression scheme as in Lemma 10, we have that with probability at least $1 - \delta$ over $S \sim \mathcal{D}^n$,

$$\mathop{\mathbf{E}}_{(x,y)\sim\mathcal{D}} \left[ \mathbb{I}\left\{ \ell(\hat{h}(x), y) > \gamma \right\} \right] \leq \frac{C}{n - n_0 \log(n)} \left( n_0 \log^2 n + \log(1/\delta) \right) ,$$

which means that for large enough $n$,

$$\mathop{\mathbf{E}}_{(x,y)\sim\mathcal{D}} \left[ \mathbb{I}\left\{ \ell(\hat{h}(x), y) > \gamma \right\} \right] \leq O \left( \frac{n_0 \log^2 n}{n} + \frac{\log(1/\delta)}{n} \right) .$$

Thus,

$$\mathop{\mathbf{Pr}}_{(x,y)\sim\mathcal{D}} \left[ \ell(\hat{h}(x), y) > \gamma \right] \leq O \left( \frac{n_0 \log^2 n}{n} + \frac{\log(1/\delta)}{n} \right) .$$

Hence, we can see that

$$\mathcal{M}(\mathcal{H}; \varepsilon, \delta, \gamma) \leq O \left( \frac{\mathbb{D}_\gamma^{\mathrm{OIG}}(\mathcal{H})}{\varepsilon} \log^2 \frac{\mathbb{D}_\gamma^{\mathrm{OIG}}(\mathcal{H})}{\varepsilon} + \frac{1}{\varepsilon} \log \frac{1}{\delta} \right) .$$

$\square$

# D $\quad \gamma$-DS Dimension and Learnability

In this section, we will show that finiteness of $\gamma$-DS dimension is necessary for PAC learning in the realizable case.

**Theorem 3.** *Let* $\mathcal{H} \subseteq [0,1]^{\mathcal{X}}, \varepsilon, \delta, \gamma \in (0,1)^3$. *Then,*

$$\mathcal{M}(\mathcal{H}; \varepsilon, \delta, \gamma) \geq \Omega \left( \frac{\mathbb{D}_{2\gamma}^{\mathrm{DS}}(\mathcal{H}) + \log(1/\delta)}{\varepsilon} \right) .$$

*Proof.* Let $d = \mathbb{D}_{2\gamma}^{\mathrm{OIG}}(\mathcal{H})$. Then, there exists some $S = (x_1, \ldots, x_d) \in \mathcal{X}^d$ such that $\mathcal{H}|_S$ contains a $2\gamma$-pseudo-cube, which we call $\mathcal{H}'$. By the definition of the scaled pseudo-cube, $\forall h \in \mathcal{H}', i \in [d]$, there is exactly one $h' \in \mathcal{H}'$ such that $h(x_j) = h'(x_j), j \neq i$, and $\ell(h(x_i), h'(x_i)) > 2\gamma$. We pick the target function $h^\star$ uniformly at random among the hypotheses of $\mathcal{H}'$ and we set the marginal distribution $\mathcal{D}_{\mathcal{X}}$ of $\mathcal{D}$ as follows

$$\mathbf{Pr}[x_1] = 1 - 2\varepsilon, \quad \mathbf{Pr}[x_i] = 2\varepsilon/(d-1), \ \forall i \in \{2, ..., d\} .$$

Consider $m$ samples $\{(z_i, h^\star(z_i)\}_{i \in [m]}$ drawn i.i.d. from $\mathcal{D}$. Let us take $m \leq \frac{d-1}{6\varepsilon}$. Then, the sample will include at most $(d-1)/2$ examples which are not $x_1$ with probability $1/100$, using Chernoff's bound. Let us call this event $E$. Conditioned on $E$, the posterior distribution of the unobserved points is uniform among the vertices of the $d/2$-dimensional $2\gamma$-pseudo-cube. Thus, if the test point $x$ falls among the unobserved points, the learner will make a $\gamma$-mistake with probability at least $1/2$. To see that, let $\hat{y}$ be the prediction of the learner on $x$. Since every hyperedge has size at least 2 and all the vertices that are on the hyperedge differ by at least $2\gamma$ in the direction of $x$, no matter what $\hat{y}$ is the correct label $y^\star$ is at least $\gamma$-far from it. Since $\mathbf{Pr}[E] \geq 1/100$, we can see that $\mathcal{M}_{\mathcal{A}}(\mathcal{H}; \varepsilon, \delta, \gamma) = \Omega(\frac{d}{\varepsilon})$. Moreover, by the law of total probability there must exist a deterministic choice of the target function $h^\star$, that could depend on $\mathcal{A}$, which satisfies the lower bound. For the other part of the lower bound, notices the probability that the sample will only contain $x_1$ is

$(1 - 2\varepsilon)^m \geq e^{-4\varepsilon m}$ which is greater that $\delta$ whenever $m \leq \log(1/\delta)/(4\varepsilon)$. This implies that the $\gamma$-cut-off sample complexity is lower bounded by

$$\max\left\{C_1 \cdot \frac{d}{\varepsilon}, C_2 \cdot \frac{\log(1/\delta)}{\varepsilon}\right\} = C_0 \cdot \frac{\mathbb{D}_{2\gamma}^{\mathrm{DS}}(\mathcal{H}) + \log(1/\delta)}{\varepsilon} .$$

Thus $\mathcal{M}_{\mathcal{A}}(\mathcal{H}; \varepsilon, \delta, \gamma)$, satisfies the desired bound when the dimension is finite. Finally, it remains to claim about the case where $\mathbb{D}_{2\gamma}^{\mathrm{DS}}(\mathcal{H}) = \infty$ for the given $\gamma$. We consider a sequence of $2\gamma$-DS shattered sets $S_n$ with $|S_n| = n$ and repeat the claim for the finite case. This will yield that for any $n$ the $\gamma$-cut-off sample complexity is lower bounded by $\Omega((n + \log(1/\delta))/\varepsilon)$ and this yields that $\mathcal{M}(\mathcal{H}; \varepsilon, \delta, \gamma) = \infty$. $\qquad\square$

We further conjecture that this dimension is also sufficient for PAC learning.

**Conjecture 2.** *A class $\mathcal{H} \subseteq (0, 1)^{\mathcal{X}}$ is PAC learnable in the realizable regression setting with respect to the absolute loss function if and only if $\mathbb{D}_{\gamma}^{\mathrm{DS}}(\mathcal{H}) < \infty$ for any $\gamma \in (0, 1)$.*

We believe that there must exist a modification of the approach of [BCD+22] that will be helpful in settling the above conjecture.

**Conjecture 3.** *There exists $\mathcal{H} \subseteq (0, 1)^{\mathcal{X}}$ for which $\mathbb{D}_{\gamma}^{\mathrm{Nat}}(\mathcal{H}) = 1$ for all $\gamma \in (0, 1)$ but $\mathbb{D}_{\gamma}^{\mathrm{DS}}(\mathcal{H}) < \infty$ for some $\gamma \in (0, 1)$.*

In particular, we believe that one can extend the construction of [BCD+22] (which uses various tools from algebraic topology as a black-box) and obtain a hypothesis class $\mathcal{H} \subseteq [0, 1]^{\mathcal{X}}$ that has $\gamma$-Natarajan dimension 1 but is not PAC learnable (it will have infinite $\gamma$-DS dimension). This construction though is not immediate and requires new ideas related to the works of [JŚ03, Osa13]

## E   Online Realizable Regression

In this section, we present our results regarding online realizable regression. The next result resolves an open question of [DG22]. It provides an online learner with optimal (off by a factor of 2) cumulative loss in realizable regression.

**Theorem 4** (Optimal Cumulative Loss). *Let $\mathcal{H} \subseteq [0, 1]^{\mathcal{X}}$ and $\varepsilon > 0$. Then, there exists a deterministic algorithm (Algorithm 3) whose cumulative loss in the realizable setting is bounded by $\mathbb{D}^{\mathrm{onl}}(\mathcal{H}) + \varepsilon$. Conversely, for any $\varepsilon > 0$, every deterministic algorithm in the realizable setting incurs loss at least $\mathbb{D}^{\mathrm{onl}}(\mathcal{H})/2 - \varepsilon$.*

---

**Algorithm 3** Scaled SOA

---

**Parameters:** $\{\varepsilon_t\}_{t \in \mathbb{N}}$.
**Initialize** $V^{(1)} = \mathcal{H}$.
For $t = 1, \ldots$:

1. Receive $x_t \in \mathcal{X}$.

2. For every $y \in [0, 1]$, let $V_{(x_t, y)}^{(t)} = \{h \in V^{(t)} : h(x_t) = y\}$.

3. Let $\widehat{y}_t$ be an arbitrary label such that

$$\mathbb{D}^{\mathrm{onl}}\left(V_{(x_t, \widehat{y}_t)}^{(t)}\right) \geq \sup_{y'} \mathbb{D}^{\mathrm{onl}}\left(V_{(x_t, y')}^{(t)}\right) - \varepsilon_t .$$

4. Predict $\widehat{y}_t$.

5. Receive the true label $y_t^{\star}$ and incur loss $\ell(\widehat{y}_t, y_t^{\star})$.

6. Update $V^{(t+1)} = \{h \in V^{(t)} : h(x_t) = y_t^{\star}\}$.

---

*Proof.* Let us begin with the upper bound. Assume that $\mathbb{D}^{\mathrm{onl}}(\mathcal{H}) < \infty$. Suppose we are predicting on the $t$-th point in the sequence and let $V^{(t)}$ be the version space so far, i.e., $V^{(t)} =$

$\{h \in \mathcal{H} : \forall \tau \in [t-1], h(x_\tau) = y_\tau\}$. Let $x_t$ be the next point to predict on. For each label $y \in \mathbb{R}$, let $V^{(t)}_{(x_t,y)} = \{h \in V^{(t)} : h(x_t) = y\}$. From the definition of the dimension $\mathbb{D}^{\mathrm{onl}}$, we know that for all $y, y' \in \mathbb{R}$ such that $V^{(t)}_{(x_t,y)}, V^{(t)}_{(x_t,y')} \neq \emptyset$,

$$\mathbb{D}^{\mathrm{onl}}(V^t) \geq \ell(y, y') + \min\left\{\mathbb{D}^{\mathrm{onl}}\left(V^{(t)}_{(x_t,y)}\right), \mathbb{D}^{\mathrm{onl}}\left(V^{(t)}_{(x_t,y')}\right)\right\}.$$

Let $\widehat{y}_t$ be an arbitrary label with $\mathbb{D}^{\mathrm{onl}}\left(V^{(t)}_{(x_t,\widehat{y}_t)}\right) \geq \sup_{y'} \mathbb{D}^{\mathrm{onl}}\left(V^{(t)}_{(x_t,y')}\right) - \varepsilon_t$, where $\varepsilon_t$ is some sequence shrinking arbitrarily quickly in the number of rounds $t$. The learner predicts $\widehat{y}_t$. Assume that the adversary picks $y_t^\star$ as the true label and, so, the learner incurs loss $\ell(\widehat{y}_t, y_t^\star)$ at round $t$. Then, the updated version space $V^{(t)}_{(x_t,y_t^\star)}$ has

$$\mathbb{D}^{\mathrm{onl}}\left(V^{(t)}_{(x_t,y_t^\star)}\right) \leq \sup_{y'} \mathbb{D}^{\mathrm{onl}}\left(V^{(t)}_{(x_t,y')}\right) \leq \mathbb{D}^{\mathrm{onl}}\left(V^{(t)}_{(x_t,\widehat{y}_t)}\right) + \varepsilon_t,$$

which implies

$$\min\left\{\mathbb{D}^{\mathrm{onl}}\left(V^{(t)}_{(x_t,\widehat{y}_t)}\right), \mathbb{D}^{\mathrm{onl}}\left(V^{(t)}_{(x_t,y_t^\star)}\right)\right\} \geq \mathbb{D}^{\mathrm{onl}}\left(V^{(t)}_{(x,y_t^\star)}\right) - \varepsilon_t.$$

This gives that

$$\mathbb{D}^{\mathrm{onl}}(V^{(t)}) \geq \ell(\widehat{y}_t, y_t^*) + \min\left\{\mathbb{D}^{\mathrm{onl}}\left(V^{(t)}_{(x_t,\widehat{y}_t)}\right), \mathbb{D}^{\mathrm{onl}}\left(V^{(t)}_{(x_t,y_t^*)}\right)\right\}$$

$$\geq \ell(\widehat{y}_t, y_t^*) + \mathbb{D}^{\mathrm{onl}}\left(V^{(t)}_{(x_t,y_t^*)}\right) - \varepsilon_t,$$

and, by re-arranging,

$$\mathbb{D}^{\mathrm{onl}}\left(V^{(t)}_{(x_t,y_t^*)}\right) \leq \mathbb{D}^{\mathrm{onl}}(V^{(t)}) - \ell(\widehat{y}_t, y_t^*) + \varepsilon_t. \tag{1}$$

So every round reduces the dimension by at least the magnitude of the loss (minus $\varepsilon_t$). Notice that $\mathbb{D}^{\mathrm{onl}}(V^{(t+1)}) = \mathbb{D}^{\mathrm{onl}}\left(V^{(t)}_{(x_t,y_t^\star)}\right)$. Thus, by choosing the $\{\varepsilon_t\}_{t\in\mathbb{N}}$ sequence such that

$$\sum_t \varepsilon_t \leq \varepsilon',$$

and summing up Equation (1) over all $t \in \mathbb{N}$, we get a cumulative loss bound

$$\sum_t \ell(\widehat{y}_t, y_t^\star) \leq \mathbb{D}^{\mathrm{onl}}(\mathcal{H}) + \varepsilon'.$$

Hence, we see that by taking the limit as $\varepsilon'$ goes to 0 shows that the cumulative loss is upper bounded by $\mathbb{D}^{\mathrm{onl}}(\mathcal{H})$. This analysis shows that Algorithm 3 achieves the cumulative loss bound $\mathbb{D}^{\mathrm{onl}}(\mathcal{H}) + \varepsilon'$, for arbitrarily small $\varepsilon' > 0$.

Let us continue with the lower bound. For any $\varepsilon > 0$, we are going to prove that any deterministic learner must incur cumulative loss at least $\mathbb{D}^{\mathrm{onl}}(\mathcal{H})/2 - \varepsilon$. By the definition of $\mathbb{D}^{\mathrm{onl}}(\mathcal{H})$, for any $\varepsilon > 0$, there exists a tree $T_\varepsilon$ such that, for every path $\boldsymbol{y}$,

$$\sum_{i=1}^{\infty} \gamma_{\boldsymbol{y}\leq i} \geq \mathbb{D}^{\mathrm{onl}}(\mathcal{H}) - 2\varepsilon,$$

i.e., the sum of the gaps across the path is at least $\mathbb{D}^{\mathrm{onl}}(\mathcal{H}) - 2\varepsilon$. The strategy of the adversary is the following: in the first round, she presents the learner with the instance $x_1 = x_\emptyset$. Then, no matter what label $\hat{y}_1$ the learner picks, the adversary can choose the label $y_1^\star$ so that $|\hat{y}_1 - y_1^\star| \geq \gamma_\emptyset/2$. The adversary can keep picking the instances $x_t$ based on the induced path of the choices of the true labels $\{y_\tau^\star\}_{\tau<t}$ and the loss of the learner in every round $t$ is at least $\gamma_{\boldsymbol{y}\leq t}/2$. Thus, summing up over all the rounds as $T \to \infty$, we see that the total loss of the learner is at least

$$\frac{\mathbb{D}^{\mathrm{onl}}(\mathcal{H})}{2} - \varepsilon.$$

$\square$

**Remark 2** (Randomized Online Learners). *We highlight that, unlike the setting of realizable online classification, in the case of realizable online regression randomization does not seem to help the learner (see also [FHMM23]). In particular, the lower bound of $\frac{\mathbb{D}^{\mathrm{onl}}(\mathcal{H})}{2} - \varepsilon$ holds even for randomized learners. To see it, notice that for all distributions over $\mathcal{D}$ over $[0,1]$ it holds that*

$$\max_{c_1,c_2}\left\{ \mathop{\mathbf{E}}_{X\sim\mathcal{D}}[\ell(X,c_1)], \mathop{\mathbf{E}}_{X\sim\mathcal{D}}[\ell(X,c_2)]\right\} \geq \ell(c_1,c_2)/2\,.$$

**Example 2** (Sequential Complexity Measures). *Sequential fat-shattering dimension and sequential covering numbers are two standard combinatorial measures for regression in online settings [RST15b, RST15a]. Note that Example 1 can be learned with 1 sample even in the online realizable setting. Hence, Example 1 shows that sequential fat-shattering dimension fails to characterize online realizable regression (since this dimension is at least as large as fat-shattering dimension which is infinite in this example). Moreover, we know that sequential covering numbers and sequential fat-shattering dimension are of the same order of magnitude and so they are also infinite in the case of Example 1.*

## F   Dimension and Finite Character Property

[BDHM$^+$19] gave a formal definition of the notion of "dimension" or "complexity measure", that all previously proposed dimensions in statistical learning theory comply with. In addition to characterizing learnability, a dimension should satisfy the finite character property:

**Definition 17** (Finite Character [BDHM$^+$19]). *A dimension characterizing learnability can be abstracted as a function $F$ that maps a class $\mathcal{H}$ to $\mathbb{N}\cup\{\infty\}$ and satisfies the finite character property: for every $d \in \mathbb{N}$ and $\mathcal{H}$, the statement "$F(\mathcal{H}) \geq d$" can be demonstrated by a finite set $X \subseteq \mathcal{X}$ of domain points, and a finite set of hypotheses $H \subseteq \mathcal{H}$. That is, "$F(\mathcal{H}) \geq d$" is equivalent to the existence of a bounded first order formula $\phi(\mathcal{X},\mathcal{H})$ in which all the quantifiers are of the form: $\exists x \in \mathcal{X}, \forall x \in \mathcal{X}$ or $\exists h \in \mathcal{H}, \forall h \in \mathcal{H}$.*

**Claim 1.** *The scaled one-inclusion graph dimension $\mathbb{D}^{\mathrm{OIG}}_\gamma(\mathcal{H})$ satisfies the finite character property.*

*Proof.* To demonstrate that $\mathbb{D}^{\mathrm{OIG}}(\mathcal{H}) \geq d$, it suffices to find a set $S$ of $n$ domain points and present a finite subgraph $G = (V,E)$ of the one-inclusion hypergraph induced by $S$ where every orientation $\sigma : E \to V$ has out-degree at least $n/3$. Note that $V$ is, by definition, a finite collection of datasets realizable by $\mathcal{H}$ and so this means that we can demonstrate that $\mathbb{D}^{\mathrm{OIG}}(\mathcal{H}) \geq d$ with a finite set of domain points and a finite set of hypotheses. $\qquad\square$

## G   Examples for Scaled Graph Dimension

These examples are adaptations from [DSS14, DSBDSS15].

**Example 3** (Large Gap Between ERM Learners). *For every $d \in \mathbb{N}$, consider a domain $\mathcal{X}_d$ such that $|\mathcal{X}_d| = d$ and $\mathcal{X}_d, \mathcal{X}_{d'}$ are disjoint for $d \neq d'$. For all $d \in \mathbb{N}$, let $P(\mathcal{X}_d)$ denote the collection of all finite and co-finite[6] subsets of $\mathcal{X}_d$. Let us fix $\gamma \in (0,1)$. Consider a mapping $f : \cup_{d\in\mathbb{N}}P(\mathcal{X}_d) \to [0,1]$ such that $f(A_d) \in (\gamma,1)$ for all $d \in \mathbb{N}, A_d \in P(\mathcal{X}_d)$, and $f(A_d) \neq f(A'_{d'})$ for all $A_d \neq A'_{d'}, A_d \in P(\mathcal{X}_d), A_{d'} \in P(\mathcal{X}_{d'})$. Such a mapping exists due to the density of the reals. For any $d \in \mathbb{N}, A_d \subseteq \mathcal{X}_d$, let $h_{A_d}(x) = f(A_d) \cdot \mathbb{1}\{x \in A_d\}$ and consider the scaled first Cantor class $\mathcal{H}_{\mathcal{X}_d,\gamma} = \{h_{A_d} : A_d \in P(\mathcal{X}_d)\}$. We claim that $\mathbb{D}^{\mathrm{Nat}}_\gamma(\mathcal{H}_{\mathcal{X}_d,\gamma}) = 1$ and that $\mathbb{D}^{\mathrm{G}}_\gamma(\mathcal{H}_{\mathcal{X}_d,\gamma}) = |\mathcal{X}_d| = d$ since one can use $f_\emptyset$ for the $\gamma$-graph shattering. Consider the following two ERM learners for the scaled first Cantor class $\mathcal{H}_{\mathcal{X}_d,\gamma}$:*

1. *Whenever a sample of the form $S = \{(x_i,0)\}_{i\in[n]}$ is observed, the first algorithm outputs $h_{\cup\{x_i\}^c_{i\in[n]}}$ which minimizes the empirical error. If the sample contains a non-zero element, the ERM learner identifies the correct hypothesis. The sample complexity of PAC learning is $\Omega(d)$.*

2. *The second algorithm either returns the all-zero function or identifies the correct hypothesis if the sample contains a non-zero label. This is a good ERM learner $\mathcal{A}^{ERM}_{good}$ with sample complexity $m(\varepsilon,\delta) = \frac{1}{\varepsilon}\log\left(\frac{1}{\delta}\right).$*

---

[6]A set $S \subseteq \mathcal{X}_d$ is co-finite if its complement $S^c$ is finite.

The construction that illustrates the poor performance of the first learner is exactly the same as in the proof of the lower bound of Theorem 1. The second part of the example is formally shown in Claim 2, which follows.

**Claim 2** (Good ERM Learner). *Let $\varepsilon, \delta \in (0,1)^2$. Then, the good ERM learner of Example 3 has sample complexity $\mathcal{M}(\varepsilon, \delta) = \frac{1}{\varepsilon} \log\left(\frac{1}{\delta}\right)$.*

*Proof.* Let $d \in \mathbb{N}$, $\mathcal{D}_{\mathcal{X}_d}$ be a distribution over $\mathcal{X}_d$ and $h_{A_d^\star}$ be the labeling function. Consider a sample $S$ of length $m$. If the learner observes a value that is different from $0$ among the labels in $S$, then it will be able to infer $h_{A_d^\star}$ and incur $0$ error. On the other hand, if the learner returns the all zero function its error can be bounded as

$$\mathop{\mathbf{E}}_{x \sim \mathcal{D}_{\mathcal{X}_d}} \left[\ell(h_\emptyset(x), h_{A_d^\star}(x))\right] \leq \mathop{\mathbf{Pr}}_{x \sim \mathcal{D}_{\mathcal{X}_d}} \left[x \in A_d^\star\right].$$

Since in all the $m = \frac{1}{\varepsilon} \log\left(\frac{1}{\delta}\right)$ draws of the training set $S$ there were no elements from $A_d^\star$ we can see that, with probability at least $1 - \delta$ over the draws of $S$ it holds that

$$\mathop{\mathbf{Pr}}_{x \sim \mathcal{D}_{\mathcal{X}_d}} \left[x \in A_d^\star\right] \leq \varepsilon.$$

Thus, the algorithm satisfies the desired guarantees. $\square$

The next example shows that no proper algorithm can be optimal in the realizable regression setting.

**Example 4** (No Optimal PAC Learner Can be Proper). *Let $\mathcal{X}_d$ contain $d$ elements and let $\gamma \in (0,1)$. Consider the subclass of the scaled first Cantor class (see Example 3) with $\mathcal{H}'_{d,\gamma} = \{h_A : A \in P(\mathcal{X}_d), |A| = \lfloor d/2 \rfloor\}$. First, since this class is contained in the scaled first Cantor class, we can employ the good ERM and learn it. However, this learner is improper since $h_\emptyset \notin \mathcal{H}'_{d,\gamma}$. Then, no proper algorithm is able to PAC learn $\mathcal{H}'_{d,\gamma}$ using $o(d)$ examples.*

*Proof.* Suppose that an adversary chooses $h_A \in \mathcal{H}'_{d,\gamma}$ uniformly at random and consider the distribution on $\mathcal{X}_d$ which is uniform on the complement of $A$, where $|A| = O(d)$. Note that the error of every hypothesis $h_B \in \mathcal{H}'_{d,\gamma}$ is at least $\gamma |B \setminus A|/d$. Therefore, to return a hypothesis with small error, the algorithm must recover a set that is almost disjoint from $A$ and so recover $A$. However the size of $A$ implies that it cannot be done with $o(d)$ examples.

Formally, fix $x_0 \in \mathcal{X}_d$ and $\varepsilon \in (0,1)$. Let $A \subseteq \mathcal{X}_d \setminus \{x_0\}$ of size $d/2$. Let $\mathcal{D}_A$ be a distribution with mass $\mathcal{D}_A((x_0, h_A(x_0))) = 1 - 16\varepsilon$ and is uniform on the points $\{(x, h_A(x)) : x \in A^c\}$, where $A^c$ is the complement of $A$ (without $x_0$).

Consider a proper learning algorithm $\mathcal{A}$. We will show that there is some algorithm-dependent set $A$, so that when $\mathcal{A}$ is run on $\mathcal{D}_A$ with $m = O(d/\varepsilon)$, it outputs a hypothesis with error at least $\gamma$ with constant probability.

Pick $A$ uniformly at random from all sets of size $d/2$ of $\mathcal{X}_d \setminus \{x_0\}$. Let $Z$ be the random variable that counts the number of samples in the $m$ draws from $\mathcal{D}_A$ that are not $(x_0, h_A(x_0))$. Standard concentration bounds imply that with probability at least $1/2$, the number of points from $(\mathcal{X}_d \setminus \{x_0\}) \setminus A$ is at most $d/4$. Conditioning on this event, $A$ is a uniformly chosen random set of size $d/2$ that is chosen uniformly from all subsets of a set $\mathcal{X}' \subset \mathcal{X}_d$ with $|\mathcal{X}'| \geq 3d/4$ (these points are not present in the sample). Now assume that the learner returns a hypothesis $h_B$, where $B$ is a subset of size $d/2$. Note that $\mathbf{E}[|B \setminus A|] \geq d/6$. Hence there exists a set $A$ such that with probability $1/2$, it holds that $|B \setminus A| \geq d/6$. This means that $\mathcal{A}$ incurs a loss of at least $\gamma$ on all points in $B \setminus A$ and the mass of each such point is $\Omega(\varepsilon/d)$. Hence, in total, the learner will incur a loss of order $\gamma \cdot \varepsilon$. $\square$

# H   Extension to More General Loss Functions

Our results can be extended to loss functions that satisfy approximate pseudo-metric axioms (see e.g., [HKLM22, CKW08]). The main difference from metric losses is that we allow an approximate triangle inequality instead of a strict inequality. Many natural loss functions are captured by this definition, such as the well-studied $\ell_p$ losses for the regression setting. Abstractly, in this context, the

label space[7] is an abstract non-empty set $\mathcal{Y}$, equipped with a general loss function $\ell : \mathcal{Y}^2 \to \mathbb{R}_{\geq 0}$ satisfying the following property.

**Definition 18** (Approximate Pseudo-Metric). *For $c \geq 1$, a loss function $\ell : \mathcal{Y}^2 \to \mathbb{R}_{\geq 0}$ is c-approximate pseudo-metric if (i) $\ell(x, x) = 0$ for any $x \in \mathcal{Y}$, (ii) $\ell(x, y) = \ell(y, x)$ for any $x, y \in \mathcal{Y}$, and, (iii) $\ell$ satisfies a c-approximate triangle inequality; for any $y_1, y_2, y_3 \in \mathcal{Y}$, it holds that $\ell(y_1, y_2) \leq c(\ell(y_1, y_3) + \ell(y_2, y_3))$.*

Furthermore, note that all dimensions for $\mathcal{H}$, $\mathbb{D}_\gamma^{\mathrm{G}}(\mathcal{H})$, $\mathbb{D}_\gamma^{\mathrm{OIG}}(\mathcal{H})$, $\mathbb{D}_\gamma^{\mathrm{DS}}(\mathcal{H})$, and $\mathbb{D}_\gamma^{\mathrm{onl}}(\mathcal{H})$ are defined for loss functions satisfying Definition 18.

Next, we provide extensions of our main results for approximate pseudo-metric losses and provide proof sketches for the extensions.

**ERM Learnability for Approximate Pseudo-Metrics.**    For ERM learnability and losses satisfying Definition 18, we can obtain the next result.

**Theorem 5.** *Let $\ell$ be a loss function satisfying Definition 18. Then for every class $\mathcal{H} \subseteq \mathcal{Y}^\mathcal{X}$, $\mathcal{H}$ is learnable by any ERM in the realizable PAC regression setting under $\ell$ if and only if $\mathbb{D}_\gamma^{\mathrm{G}}(\mathcal{H}) < \infty$ for all $\gamma \in (0, 1)$.*

The proof of the upper bound and the lower bound follow in the exact same way as with the absolute loss.

**PAC Learnability for Approximate Pseudo-Metrics.**    As for PAC learning with approximate pseudo-metric losses, we can derive the next statement.

**Theorem 6.** *Let $\ell$ be a loss function satisfying Definition 18. Then every class $\mathcal{H} \subseteq \mathcal{Y}^\mathcal{X}$ is PAC learnable in the realizable PAC regression setting under $\ell$ if and only if $\mathbb{D}_\gamma^{\mathrm{OIG}}(\mathcal{H}) < \infty$ for any $\gamma \in (0, 1)$.*

*Proof Sketch.* We can generalize the upper bound in Theorem 2 for the scaled OIG dimension as follows. One of the ingredients of the proof for the absolute loss is to construct a sample compression scheme through the median boosting algorithm (cf. Algorithm 2). While the multiplicative update rule is defined for any loss function, the median aggregation is no longer the right aggregation for arbitrary (approximate) pseudo-metrics. However, for each such loss function, there exists an aggregation such that the output value of the ensemble is within some cutoff value from the true label for each example in the training set, which means that we have a sample compression scheme for some cutoff loss. In particular, we show that by using weak learners with cutoff parameter $\gamma/(2c)$, where $c$ is the approximation level of the triangle inequality, the aggregation of the base learners can be expressed as a sample compression scheme for cutoff loss with parameter $\gamma$.

Indeed, running the boosting algorithm with $(\gamma/(2c), 1/6)$-weak learners yields a set $h_1, \dots, h_N$ of weak predictors, with the property that for each training example $(x, y)$, at least $2/3$ of the functions $h_i$ (as weighted by coefficients $\alpha_i$), $1 \leq i \leq N$, satisfy $\ell(h_i(x), y) \leq \gamma/(2c)$. For any $x$, let $\hat{h}(x)$ be a value in $\mathcal{Y}$ such that at least $2/3$ of $h_i$ (as weighted by $\alpha_i$), $1 \leq i \leq N$, satisfy $\ell(h_i(x), \hat{h}(x)) \leq \gamma/(2c)$, if such a value exists, and otherwise $\hat{h}(x)$ is an arbitrary value in $\mathcal{Y}$. In particular, note that on the training examples $(x, y)$, the label $y$ satisfies this property, and hence $\hat{h}(x)$ is defined by the first case. Thus, for any training example, there exists $h_i$ (indeed, at least $2/3$ of them) such that *both* $\ell(h_i(x), y) \leq \gamma/(2c)$ and $\ell(h_i(x), \hat{h}(x)) \leq \gamma/(2c)$ are satisfied, and therefore we have

$$\ell(\hat{h}(x), y) \leq c(\ell(\hat{h}(x), h_i(x)) + \ell(h_i(x), y)) \leq \gamma.$$

This function $\hat{h}$ can be expressed as a sample compression scheme of size $O\left(\mathbb{D}_{\gamma/(2c)}^{\mathrm{OIG}}(\mathcal{H}) \log(m)\right)$ for cutoff loss with parameter $\gamma$: namely, it is purely defined by the $h_i$ functions, where each $h_i$ is

---

[7]We would like to mention that, in general, we do not require that the label space is bounded. In contrast, we have to assume that the loss function takes values in a bounded space. This is actually necessary since having an unbounded loss in the regression task would potentially make the learning task impossible. For instance, having some fixed accuracy goal, one could construct a learning instance (distribution over labeled examples) that would make estimation with that level of accuracy trivially impossible.

specified by $O\left(\mathbb{D}^{\mathrm{OIG}}_{\gamma/(2c)}(\mathcal{H})\right)$ training examples, and we have $N = O(\log(m))$ such functions, and $\hat{h}$ satisfies $\ell(\hat{h}(x), y) \leq \gamma$ for all $m$ training examples $(x, y)$. Thus, by standard generalization bounds for sample compression, we get an upper bound that scales with $\tilde{O}\left(\mathbb{D}^{\mathrm{OIG}}_{\gamma/(2c)}(\mathcal{H})\frac{1}{m}\right)$ for the cutoff loss with parameter $\gamma$, and hence by Markov's inequality, an upper bound

$$\mathbf{E}[\ell(\hat{h}(x), y)] = \tilde{O}\left(\mathbb{D}^{\mathrm{OIG}}_{\gamma/(2c)}(\mathcal{H})\frac{1}{m\gamma}\right).$$

We next deal with the lower bound. For the absolute loss, we scale the dimension by $2\gamma$ instead of $\gamma$ since for any two possible labels $y_1, y_2$ the learner can predict some intermediate point, and we want to make sure that the prediction will be either $\gamma$ far from $y_1$ or $y_2$. For an approximate pseudo-metric, we should take instead $2c\gamma$ in order to ensure that the prediction is $\gamma$ far, which means that the lower bounds in Theorem 2 hold with a scale of $2c\gamma$. □

**Online Learnability for Approximate Pseudo-Metrics.**     Finally, we present the more general statement for online learning.

**Theorem 7.** *Let $\ell$ be a loss function satisfying Definition 18 with parameter $c \geq 1$. Let $\mathcal{H} \subseteq \mathcal{Y}^{\mathcal{X}}$ and $\varepsilon > 0$. Then, there exists a deterministic algorithm whose cumulative loss in the realizable setting is bounded by $\mathbb{D}^{\mathrm{onl}}(\mathcal{H}) + \varepsilon$. Conversely, for any $\varepsilon > 0$, every deterministic algorithm in the realizable setting incurs loss at least $\mathbb{D}^{\mathrm{onl}}(\mathcal{H})/(2c) - \varepsilon$.*

*Proof Sketch.* The upper bound of Theorem 4 works for any loss function. Recall the proof idea; in every round $t$ there is some $\widehat{y}_t \in \mathcal{Y}$ the learner can predict such that no matter what the adversary picks as the true label $y_t^\star$, the online dimension of the version space at round $t$, i.e, $V = \{h \in \mathcal{H} : h(x_\tau) = y_\tau^\star, 1 \leq \tau \leq t\}$, decreases by $\ell(y_t^\star, \widehat{y}_t)$, minus some shrinking number $\epsilon_t$ that we can choose as a parameter. Therefore we get that the sum of losses is bounded by the online dimension and the sum of $\epsilon_t$ that we can choose to be arbitrarily small.

The lower bound for online learning in Theorem 4 would be $\mathbb{D}^{\mathrm{onl}}(\mathcal{H})/(2c) - \varepsilon$, for any $\epsilon > 0$, since the adversary can force a loss of $\gamma_{\boldsymbol{y}_{\leq t}}/(2c)$ in every round $t$, where $\gamma_{\boldsymbol{y}_{\leq t}}$ is the sum of the gaps across the path $\boldsymbol{y}$.

□

