# OpenReview forum: "Optimal Learners for Realizable Regression: PAC Learning and Online Learning"
_NeurIPS.cc/2023/Conference — NeurIPS 2023 oral_

### Official Review · Reviewer_JHXn · 2023-07-06

**Soundness:** 4 excellent
**Presentation:** 4 excellent
**Contribution:** 4 excellent
**Rating:** 8
**Confidence:** 3

**Summary:**

This paper studies the statistical complexity of realizable regression in the PAC learning and online learning setups. The main results are the following combinatorial conditions that characterize the PAC and online learnability:
- PAC learnability by (worst-case) ERM learner is equivalent to having a finite $\gamma$-graph dimension for all $\gamma \in (0, 1)$.
- PAC learnability is equivalent to the finiteness of $\gamma$-one-inclusion graph dimension for all $\gamma \in (0, 1)$.
- The minimax cumulative loss in online learning is characterized (up to a constant factor) by the online dimension.
The combinatorial dimensions are above are newly introduced in the paper. The authors also conjectured that the DS dimension in the literature also characterizes PAC learnability.

In addition, the paper provides several other examples that shed light on the landscape between learnability, uniform convergence, and other complexity measures of the hypothesis class (Figure 1).

**Strengths:**

This paper studies a fundamental problem in learning theory, which has, surprisingly, been left open for several decades. The results are strong and comprehensive, and the authors did a great job in introducing the prior results and presenting the high-level roadmaps behind the technical proofs.

**Weaknesses:**

My only complaint is on the short conclusion and a lack of discussion on future directions (apart from the obvious one of proving Conjecture 1).

**Questions:**

Regarding Conjecture 1:
- Could you elaborate on the obstacle that prevents the approach of [BCD+22] to be applied towards the regression setting?
- Are there any evidence/heuristic arguments that support the conjecture? Are there interesting assumptions under which finite $\gamma$-DS dimension implies learnability?

**Limitations:**

This is a theory paper and its limitations lie in the assumptions on which the validity of the results rely, including the realizability assumption and the focus on PAC and online learning. This has been formally stated in the paper, and also explicitly mentioned in the title and abstract.

---

> ### Author Rebuttal · Authors · 2023-08-08
>
> We would like to thank the Reviewer for the positive feedback on the significance and presentation of our results and the interesting questions and suggestions.
>
> > *My only complaint is on the short conclusion and a lack of discussion on future directions (apart from the obvious one of proving Conjecture 1).*
>
> In the next version of our draft, we will include a more detailed conclusion section where we will summarize the main contributions of our work and the important next steps. Another future direction, that is not directly related to Conjecture 1, is  to better understand the gap between the fat-shattering dimension and the OIG-based dimension. In particular, it would be interesting to come up with examples of ``natural’’ hypothesis classes, other than the one we provide in Example 1, which witness the fact that the fat-shattering dimension does not characterize learnability.
>
> >*Could you elaborate on the obstacle that prevents the approach of [BCD+22] to be applied towards the regression setting?*
>
> In [Brukhim et al., 2022], the authors start by showing that if the DS dimension of $H$ is bounded by $d$, then using the OIG algorithm they can derive a learner that has error at most $d/(d+1)$. Notice that this learner is very weak, but has non-trivial guarantees. Subsequently, they use non-trivial arguments that go through list-PAC learning and sample-compression schemes to boost this very weak learner. This step is crucial for the multiclass setting since it reduces nontrivially the infinite label space. However, in the realizable regression setting, it is trivial to derive a learner that has error at most $1/2$. Indeed, if we focus on the $\ell_1$ loss and $Y = [0,1]$, by always predicting $1/2$ we can design such a learner. To the best of our knowledge, using the definition of the $\gamma$-DS dimension in a similar way as in [Brukhim et al., 2022] does not result in a non-trivial learner in the regression setting. This is the main and crucial difference between classification and regression.
>
> >*Are there any evidence/heuristic arguments that support the conjecture? Are there interesting assumptions under which finite  gamma-DS dimension implies learnability?*
>
> Let us first elaborate on the connection between OIG and the DS dimension. We will focus on the multiclass setting, studied in  [Brukhim et al., 2022]. Our Conjecture 1 essentially claims that this connection extends to the realizable regression task. Interestingly, there is some notion of “duality” between one-inclusion graph algorithms and pseudo-cubes (the combinatorial objects that define the DS dimension). In particular, Lemmas 12 and 13 in [Brukhim et al. 2022] show that there is a certain duality between orientations of OIG and the DS dimension. In particular, let us consider a class $H$ with DS dimension $d$. Then intuitively even if the algorithm is given $d$ labeled examples, then any orientation of the OIG will have large out-degree (which means that it will be a bad learner). More to that, if the algorithm is given $d+1$ labeled examples, then there exists a good orientation (one with small out-degree) which implies the existence of a good learner. These intuitive statements shed light to the connection between the DS dimension and learnability through the OIG structure.
>
> Deriving sufficient conditions is actually an interesting question for future work, and probably easier than proving the conjecture to its full extent. The reason we believe it is true is that, similar to the multiclass classification problem, the $\gamma$-DS dimension feels more ``natural’’ than the dimensions that have been proposed in the past, and seems to be capturing the learnability problem in a tighter way. Moreover, its definition is closely related to the outdegree of the OIG (as we discussed in the previous paragraphs), which, as we have shown, is the quantity that controls learnability in this setting.

---

> > ### Comment · Reviewer_JHXn · 2023-08-14
> >
> > I would like to thank the authors for their detailed answers to my questions. I don't have further questions and my positive evaluation of the paper is unchanged.

---

### Official Review · Reviewer_dt9f · 2023-07-06

**Soundness:** 4 excellent
**Presentation:** 4 excellent
**Contribution:** 4 excellent
**Rating:** 9
**Confidence:** 4

**Summary:**

This paper develops optimal learners and characterizes learnability with new combinatorial dimensions for realizable regression (where the best predictor has zero regret) in PAC and online learning, significantly depicting the landscape of learnability in PAC/online learning.

For PAC learning, they show that:
- the PAC learnability in the realizable regression by a worst-case ERM iff the gamma graph dimension is finite
- learnability in the realizable regression is fully characterized by a finite gamma One-Inclusion dimension
- finite gamma-DS dimension is a necessary condition for PAC learnability in the realizable regression

For online learning, they devise a new combinatorial dimension, namely online dimension that is built up on the scaled Littlestone dimension. They show that the online dimension characterizes the minimax instance optimal cumulative loss up to a constant factor and design an optimal online learner.

**Strengths:**

- Significant results that complete the landscape of learnability of PAC/online learning in realizability regression


**Weaknesses:**

- None that I know (note that this problem area is not my research domain)

**Questions:**

N/A

**Limitations:**

The paper might need to discuss the limitations of the results and analysis.

---

> ### Author Rebuttal · Authors · 2023-08-08
>
> We would like to thank the Reviewer for the positive feedback on the significance of our results.
>
> > *The paper might need to discuss the limitations of the results and analysis.*
>
> We believe that the main limitation of our work is that the OIG-based dimension we propose is more complicated than the dimensions that have been proposed in the past, like the fat-shattering dimension (which, as we explain, does not characterize learnability in the realizable regression setting). Nevertheless, despite its complexity, this is the first dimension that characterizes learnability in the realizable regression setting. More to that, our work leaves as an important next step is to prove (or disprove) our conjecture that the (combinatorial and simpler) $\gamma$-DS dimension is qualitatively equivalent to the $\gamma$-OIG dimension. We will add a discussion on the limitations in the first revision of our manuscript.

---

> > ### Comment · Reviewer_dt9f · 2023-08-16
> >
> > I thank the authors for the response. After enriching myself further with the relevant literature, I think the contributions in this paper are solid on fundamental levels and add important progress in the learning theory community. I thus increased my score from 7 to 9, and my confidence from 2 to 4.

---

> > > ### Author Response · Authors · 2023-08-16
> > >
> > > We are grateful to the reviewer for taking the time to familiarize themselves further with the literature and for appreciating our contributions.

---

### Official Review · Reviewer_DqR8 · 2023-07-11

**Soundness:** 3 good
**Presentation:** 3 good
**Contribution:** 3 good
**Rating:** 6
**Confidence:** 2

**Summary:**

This paper introduce some dimensions that characterize PAC learnability for realizable regression. The authors introduce $\gamma-$ Graph dimenion which is necessary and sufficient for PAC learnability by ERM, and $\gamma-$OIG dimension which is necessary and sufficient for PAC learnability. $\gamma-$DS dimension is introduced which is necessary and conjectured to be sufficient. There are also results for online learning.

**Strengths:**

PAC learnability for realizable regression is characterized by an appropriate dimension. This seems to be an important open problem that is resolved.

**Weaknesses:**

The various dimensions are hard to understand. It would be nice to see examples. For instance, lines 188-192 were not particularly helpful to understand Definition 5, since I am not sure what it means for $\mathcal{H}$ to contain a cube. Do you mean there is a hypercube of a certain size embedded in every function in $\mathcal{H}$?

**Questions:**

Line 263, is there some typo? maybe you dont mean $\forall i$.

---

> ### Author Rebuttal · Authors · 2023-08-08
>
> We would like to thank the Reviewer for the positive and insightful feedback and questions.
>
> > *The various dimensions are hard to understand. It would be nice to see examples. For instance, lines 188-192 were not particularly helpful to understand Definition 5, since I am not sure what it means for H  to contain a cube. Do you mean there is a hypercube of a certain size embedded in every function in  H ?*
>
> Let us first give some intuition behind the definition of the fat-shattering and $\gamma$-Natarajan dimensions. The other dimensions follow in a similar manner. The crucial idea is to understand what it means to shatter a set of points in each definition. Then the associated dimension is the maximum size of a set shattered by the hypothesis class.
>
> Fat-shattering dimension is a natural way to quantify how well the function class can interpolate (with gap $\gamma$) some fixed function. Crucially this interpolation contains only inequalities (see Definition 4) and hence (at least intuitively) cannot be tight for the realizable setting, where there exists some function that exactly labels the features. Example 1 gives a natural example of a class with infinite fat-shattering dimension but which can be learned with a single sample in the realizable setting.
>
> Before explaining the $\gamma$-Natarajan dimension, let us begin with the definition of the standard Natarajan dimension [Natarajan, 1989]. We say that a set $S = \\{x_1,...,x_n\\}$ of size $n$ is Natarajan-shattered by a concept class $H \subseteq \mathcal{Y}^\mathcal{X}$ if there exist two functions $f,g : S \to \mathcal{Y}$ so that $f(i) \neq g(i)$ for any $i \in S$ and
> for any $b \in \\{0,1\\}^n$ there exists $h \in H$ such that
> $h(x_i) = f(x_i)$ if $b_i = 1$ and
> $h(x_i) = g(x_i)$ if $b_i = 0.$ Note that here we have equalities instead of inequalities (recall the fat-shattering case).
>
> From a geometric perspective (see [Brukhim et al., 2022]), this means that the space $H$ projected on the set $S$ contains the set
> $ \\{ f(x_1), g(x_1)\\} \times … \\{ f(x_n), g(x_n) \\} $.
> This set is "isomorphic" to the Boolean hypercube of size $n$ by mapping $f(x_i)$ to 1 and $g(x_i)$ to 0 for any $i \in [n]$.
> This means that the Natarajan dimension is essentially the size of the largest Boolean cube contained in $H$.
>
> The $\gamma$-Natarajan dimension is the scaled version of the above definition. The only modification is that the two functions $f,g$ map $S$ to $[0,1]$ and we require that not only $f$ and $g$ are everywhere different but we have that the distance between $f(x_i)$ and $g(x_i)$ is at least $\gamma$. Nevertheless, the geometric perspective is still the same: $\gamma$-Natarajan dimension is the size of the largest Boolean cube contained in $H$. This is what we mean after Definition 5.
>
> Examples 3 and 4 contain some examples computing $\gamma$-Graph and Natarajan dimensions. Adding more intuitive examples is an important direction for future revisions.
>
> We will make the above discussion more clear in the first revision of our work.
>
>
> > *Line 263, is there some typo? maybe you dont mean \forall i*
>
> This is indeed a typo, thanks for bringing it up. We meant to say $\forall j \in [n] \setminus \{ i \}$, not $\forall i, j$. We will fix it in the next version of our draft.

---

### Official Review · Reviewer_EXPx · 2023-07-13

**Soundness:** 4 excellent
**Presentation:** 4 excellent
**Contribution:** 4 excellent
**Rating:** 7
**Confidence:** 3

**Summary:**

This work analyzes the realizable regression and connects it with several notions of dimensions. They care about the online learning and the PAC learning. They first show that the $\gamma$-OIG dimension characterizes the PAC learning and that PAC learning requires finite $\gamma$-DS. Finally, they show that for the online regression, the authors find a dimension that characterize it. They show that this dimension is an upper bound over the cumulative loss and it is a lower bound up to some constant.

**Strengths:**

 The paper provides complexity results for regression in online learning and PAC learning. In binary classification, we have a better understanding of the complexity and how different dimensions connect. In the regression setting, we do not know a lot and this paper provides a very good understanding and nice results. The paper is well written and explains the previous work well.

**Weaknesses:**

Not a weakness, but can the authors explain why is there a requirement for bounded labels? What happens if the labels are not bounded?

**Questions:**

The question in the weaknesses section.

**Limitations:**

no limitations.

---

> ### Author Rebuttal · Authors · 2023-08-08
>
> We would like to thank the Reviewer for the positive feedback regarding the importance and the clarity of our results.
>
>
> > *Not a weakness, but can the authors explain why is there a requirement for bounded labels? What happens if the labels are not bounded?*
>
> We would like to mention that, in general, we do not require that the label space is bounded. In contrast, we have to assume that the loss function takes values in a bounded space. This is actually necessary since having an unbounded loss in the regression task would potentially make the learning task impossible. For instance, having some fixed accuracy goal, one could construct a learning instance (distribution over labeled examples) that would make estimation with that level of accuracy trivially impossible. We will clarify this point in the first revision of our work.

---

### Official Review · Reviewer_seAb · 2023-07-30

**Soundness:** 4 excellent
**Presentation:** 4 excellent
**Contribution:** 3 good
**Rating:** 8
**Confidence:** 3

**Summary:**

This paper provides combinatorial dimensions that characterize realizable regression in both batch as well as online settings. Moreover, it provides minimax optimal learner up to polylog factor in the batch setting and minimax optimal learner in the online setting.

**Strengths:**

1. The paper is well-written, easy to follow, and solves an important open problem of characterizing realizable learnability for real-valued function classes.
2. The paper uses classical ideas such as Median Boosting algorithm, sample compression schemes as well as some recent developments in PAC learning theory such as partial concept classes, OIG based dimensions, etc. Overall, the paper is technically sound and is definitely an important technical contribution to the field.
3. In online setting, the paper introduces a novel idea of summing scales along each branch of the tree and defining dimension as the sum of scales. This is a novel and useful technical tool as it provides a new way of defining dimensions that are not parametrized by a scale even though some form of scale is inherent to the problem setting.

**Weaknesses:**

Although the paper does provide a combinatorial characterization of realizable regression, I am not sure if the OIG-based dimension is very insightful. Theoretically, it is a useful abstraction as it has a finite-character property and thus the learnability of the problem can, at least technically, be determined using finitely many domain points and functions in function classes. However, the practical utility of such dimension is questionable. Can be computed for natural classes such a linear classes, Lipschitz classes, and so forth? Computing upper bounds is generally difficult even for classical dimensions like VC and fat-shattering, but the lower bounds of these dimensions are typically easy to compute for some natural classes because of simplicity of their shattering conditions. Is it also the case for this OIG based dimension?





**Questions:**

I assume that fat-shattering dimension upper bounds the OIG based dimension proposed here. Is there a combinatorial proof of this fact? Also, is there a general property of the class that guarantees that the finiteness of OIG based dimension and fat-shattering dimension co-incide?

---

> ### Author Rebuttal · Authors · 2023-08-09
>
> We would like to thank the Reviewer for the positive feedback and insightful questions.
>
> > *Although the paper does provide a combinatorial characterization of realizable regression, I am not sure if the OIG-based dimension is very insightful. Theoretically, it is a useful abstraction as it has a finite-character property and thus the learnability of the problem can, at least technically, be determined using finitely many domain points and functions in function classes. However, the practical utility of such dimension is questionable. Can be computed for natural classes such a linear classes, Lipschitz classes, and so forth? Computing upper bounds is generally difficult even for classical dimensions like VC and fat-shattering, but the lower bounds of these dimensions are typically easy to compute for some natural classes because of simplicity of their shattering conditions. Is it also the case for this OIG based dimension?*
>
> In general, we believe that it is difficult to compute the $\gamma$-OIG dimension. Nevertheless, $\gamma$-OIG dimension is the first complexity measure that tightly characterizes realizable regression. More to that, we propose the much more combinatorial $\gamma$-DS dimension which we conjecture to be the right dimension for this setting. As the reviewer suggests, it would be interesting to compute the OIG-based complexity measure for natural and useful complexity classes. For instance, for the class of linear functions $f(x) = a \cdot x$, when the features are single-dimensional then $\mathbb{D}^{\mathrm{OIG}}_\gamma = 1$ (one sample suffices; each hyperedge of the OIG is a single hypothesis) and the OIG dimension scales with the dimension of the feature space in higher dimensions. Similar analysis can be done for the affine case. Deriving bounds for other families of functions is an important yet non-trivial question.
>
> > *I assume that fat-shattering dimension upper bounds the OIG based dimension proposed here. Is there a combinatorial proof of this fact? Also, is there a general property of the class that guarantees that the finiteness of OIG based dimension and fat-shattering dimension coincide?*
>
> The work of Mandelson (2002) provides a sufficient and natural condition that implies finiteness of both measures. In particular, Section 5 of this work shows that classes that contain functions with bounded oscillation (as defined in the aforementioned paper) have finite fat-shattering dimension. This implies that the class is learnable in the agnostic setting and hence is also learnable in the realizable setting. As a result, the OIG-based dimension is also finite. So, bounded oscillations are a general property that guarantees that the finiteness of OIG-based dimension and fat-shattering dimension coincide. For the first part of the question, we are not familiar with a combinatorial proof of this statement. Investigating such connections would be an interesting direction for future work.
>
> [Mandelson, Improving the Sample Complexity Using Global Data, 2002]

---

> > ### Comment · Reviewer_seAb · 2023-08-11
> >
> > Thank you for answering my question and addressing my concern about the potential weakness. I will be eagerly following the progress on the conjecture regarding $\gamma$-$\text{DS}$.
> >
> > Overall,  the paper tackles a foundational problem of learning theory. The results are significant and the techniques are sound, so I think the paper deserves a highlight at the conference. I am happy to raise the score to 8.

---

> > > ### Author Response · Authors · 2023-08-12
> > >
> > > Thank you very much for taking the time to read our rebuttal and for appreciating our work. We will make sure to address the questions that you and the rest of the reviewers raised in the next version of our draft.

---

### Decision · Program_Chairs · 2023-09-21

**Decision:**

Accept (oral)

**Comment:**

This paper solves an open problem in the learning theory community that is of fundamental importance (i.e. the statistical complexity of realizable regression in both the PAC setting and the online setting), and both introduces novel conceptual quantities and techniques. All reviewers are very enthusiastic about the paper's contributions and supportive of an oral presentation.